# Long-term analysis of clear-sky new particle formation events and non-events in Hyytiälä

Lubna Dada[1], Pauli Paasonen[1], Tuomo Nieminen [1,2], Stephany Buenrostro Mazon[1], Jenni Kontkanen[1], Otso Peräkylä[1], Katrianne Lehtipalo [1,4], Tareq Hussein[1,5], Tuukka Petäjä[1], Veli-Matti Kerminen[1], Jaana Bäck[3], and Markku Kulmala[1]

[1]Department of Physics, University of Helsinki, P.O. Box 64, FIN-00014 Helsinki, Finland
[2]Department of Applied Physics, University of Eastern Finland, P.O. Box 1627, FI-70211 Kuopio, Finland
[3]Department of Forest Sciences, University of Helsinki, P.O. Box 27, FIN-00014 Helsinki, Finland
[4] Laboratory of Atmospheric Chemistry, Paul Scherrer Institute (PSI), 5232 Villigen PSI, Switzerland
[5]Department of Physics, the University of Jordan, Amman 11942, Jordan
*Correspondence to*: Lubna Dada (lubna.dada@helsinki.fi)

**Abstract.** New particle formation (NPF) events have been observed all around the world and are known to be a major source of atmospheric aerosol particles. Here we combine 20 years of observations in a boreal forest at the SMEAR II station (Station for Measuring Ecosystem-Atmosphere Relations) in Hyytiälä, Finland, by building on previously accumulated knowledge, and by focusing on clear-sky (non-cloudy) conditions. We first investigated the effect of cloudiness on NPF and then compared the NPF event and non-event days during clear-sky conditions. In this comparison we considered, for example, the effects of calculated particle formation rates, condensation sink, trace gas concentrations and various meteorological quantities in discriminating NPF events from non-events. The formation rate of 1.5 nm particles was calculated by using proxies for gaseous sulfuric acid and oxidized products of low volatile organic compounds, together with an empirical nucleation rate coefficient. As expected, our results indicate an increase in the frequency of NPF events under clear-sky conditions in comparison to cloudy ones. Also, focusing on clear-sky conditions enabled us to find a clear separation of many variables related to NPF. For instance, oxidized organic vapors showed higher concentration during the clear-sky NPF event days, whereas the condensation sink (CS) and some trace gases had higher concentrations during the non-event days. The calculated formation rate of 3 nm particles showed a notable difference between the NPF event and non-event days during clear-sky conditions, especially in winter and spring. For spring time, we are able to find a threshold equation for the combined values of ambient temperature and CS, (CS (s-1) > $-3.091 \times 10^{-5} \times T$ (in Kelvin) + 0.0120), above which practically no clear-sky NPF event could be observed. Finally, we present a probability distribution for the frequency of NPF events at a specific CS and temperature.

Keywords: Boreal forest, formation rate, atmospheric aerosols, aerosol dynamics, condensation sink, cloudiness parameter

## 1    Introduction

The effects of atmospheric aerosol particles on the climate system, human health and environmental interactions have raised the interest in various phenomena associated with the formation, growth and loss of these particles (Pöschl, 2005; Seinfeld and Pandis, 2012; Apte et al., 2015). While primary emissions are a very important source of atmospheric aerosol particles, especially in terms of the aerosol mass loading, the particle number concentration is greatly affected by atmospheric new particle formation (NPF). During the last couple of decades, NPF has been observed to take place almost all over the world (Kulmala et al., 2004; Zhang et al., 2011; Bianchi et al., 2016; Kontkanen et al., 2016a; Kontkanen et al., 2017). Atmospheric NPF is thought to be the dominant source of the total particle number concentration (Kulmala et al., 2016), and a major source of cloud condensation nuclei, in the global troposphere (Merikanto et al., 2009; Yu et al., 2010; Kerminen et al., 2012; Salma et al., 2016).

Understanding the NPF phenomenon requires understanding its precursors and pathways involved under different atmospheric conditions. For instance, high concentrations of low-volatility vapors result in a higher probability for NPF (Nieminen et al., 2015), whereas a high relative humidity and condensation sink tend to suppress NPF (Hyvönen et al., 2005; Nieminen et al., 2014). Recent laboratory experiments have shown the importance of sulfuric acid and low-volatile oxidized organic vapors to NPF (Metzger et al., 2010; Kirkby et al., 2011; Petäjä et al., 2011; Kulmala et al., 2013; Ehn et al., 2014; Riccobono et al., 2014). Additionally, atmospheric observations confirm the importance of these precursor vapors in the initial steps of NPF and in the further growth of newly-formed particles (Kulmala et al., 1998; Smith et al., 2005; Kerminen et al., 2010; Paasonen et al., 2010; Ahlm et al., 2012; Bzdek et al., 2014; Nieminen et al., 2014; Vakkari et al., 2015). The Station for Measuring Forest Ecosystem-Atmosphere Relations (SMEAR II), located in Hyytiälä, southern Finland, compiles almost 21 years of particle number size distribution and extensive complementary data, providing the longest size distribution time series in the world, and hence allows for robust NPF analysis which is not readily possible at other sites. The station is located in a homogenous Scots pine forest far from major pollution sources. Hyytiälä is therefore classified as a background site representative of the semi-clean northern hemisphere boreal forests.

Many studies have investigated the role of different variables in causing, enhancing or preventing new particle formation (Hyvönen et al., 2005; Baranizadeh et al., 2014; Nieminen et al., 2014). In particular, Baranizadeh et. al (2014) studied the effect of cloudiness on NPF events observed at SMEAR II in Hyytiälä. They concluded, in agreement with some other studies, that clouds tend to attenuate or interrupt NPF events (Sogacheva et al., 2008; Boulon et al., 2010; Baranizadeh et al., 2014; Nieminen et al., 2015) . In this study, we eliminated one variable that limits NPF (cloudiness), in order to provide a better insight into the other quantities related to atmospheric NPF. Based on 20 years of observations and data analysis for the SMEAR II station in Hyytiälä, we aim to i) quantify the effect of cloudiness on new particle formation frequency, ii) characterize the differences between NPF event and non-event days during clear-sky conditions iii) explore the connections between new particle formation rates calculated from precursor vapor proxies and the occurrence of NPF events, iv) formulate an equation that predicts whether a clear-sky day with specific temperature and CS is classified as an event; v) use the clear-sky data set to calculate the NPF probability distribution based on temperature and CS.

## 2 Materials and methods

### 2.1 Measurements

The data used for the analysis in this study is from the University of Helsinki SMEAR (Station for Measurement of Ecosystem –Atmosphere Relations) II station (Hari and Kulmala, 2005). The station provides long-term continuous comprehensive measurements of quantities describing atmospheric-forest-ecosystem interactions. The SMEAR II station is located in the boreal forest in Hyytiälä, southern Finland (61°51N', 24°17E', 181 m a.s.l.), 220 km NW of Helsinki. Tampere (200,000 inhabitants) is the largest city nearest to the station and is located 60 km SW of the site. Being far from major human activities and surrounded by a homogenous scots pine belt, Hyytiälä is considered as a rural background site due to the low levels of air pollutants (Asmi et al., 2011). A more detailed overview of the measurements at the station can be found in Hari and Kulmala (2005) and Nieminen et al. (2014).

In this study, the data analysis is based on four types of measurements: (i) aerosol particle number size distributions, (ii) concentration of the trace gases (CO, NO, $NO_2$, $NO_x$, $SO_2$ and $O_3$), (iii) meteorological parameters (solar radiation, temperature and relative humidity), and (iv) precursor vapor concentrations from previously-developed proxies. The collection of data started in January 1996. Trace gas concentrations are measured at 6 different heights on a 74-m-high mast (extended to 126 m in summer 2010). Gas concentrations used in this study are collected from the middle level on the mast above the forest (at 16.8 m).

The aerosol number concentration size distributions were measured with a twin-DMPS (Differential Mobility Particle Sizer) system (Aalto et al., 2001) for the size ranges 3-500 nm until year 2004 and 3-1000 nm from 2005 onwards. These data were used to classify days as NPF events and non-events following the method proposed by Dal Maso et al. (2005). The size distributions obtained from the DMPS measurements were used to calculate the condensation sink, CS, which is equal to the rate at which non-volatile vapors condense onto a pre-existing aerosol particle population (Kulmala et al., 2012).

The CO concentration is measured with one infrared light absorption analyzer (API 300EU, Teledyne Monitor Labs, Englewood, CO, USA). The NO and NOx concentrations are monitored with a chemiluminescence analyzer (TEI 42C TL, Thermo Fisher Scientific, Waltham, MA, USA). The $NO_2$ concentration is calculated from the difference $NO_x–NO$. The detection limit is about 0.05 ppb. $SO_2$ measurements are made through a UV-fluorescence analyzer (TEI 43 CTL, Thermo Fisher Scientific, Waltham, MA, USA) that has a detection limit of 0.1 ppb. The $O_3$ concentration is measured with an UV light absorption analyzer (TEI 49C, Thermo Fisher Scientific, Waltham, MA, USA) that has a detection limit of about 1 ppb. The data for trace gases are available as 30-minute arithmetic means.

Solar radiation in the wavelengths of UV-B (280 – 320nm) and global radiation (0.30 - 4.8 μm) are monitored using pyranometers (SL 501A UVB, Solar Light, Philadelphia, PA, USA; Reeman TP 3, Astrodata, Tõravere, Tartumaa, Estonia until June 2008, and Middleton Solar SK08, Middleton Solar, Yarraville, Australia since June 2008) above the forest at 18 m. The air temperature is measured with 4-wired PT-100 sensors, and the relative humidity (in percent) is measured

with relative humidity sensors (Rotronic Hygromet MP102H with Hygroclip HC2-S3, Rotronic AG, Bassersdorf,
Switzerland). These data are provided as 30-minute averages.

**2.2    Data analysis**
**2.2.1    New particle formation events classification**

Formation of new aerosol particles in Hyytiälä is typically observed in the time window of several hours around noon,
while this phenomenon seems to be rare during nighttime (Junninen et al., 2008; Buenrostro Mazon et al., 2016).
Accordingly, aerosol number size distributions data from the DMPS measurements at around this time window are used
for classifying individual days as new particle formation event or non-event days. The classification follows the guidelines
presented by Kulmala et al. (2012), and the procedure presented in Dal Maso et al. (2005). The latter uses a decision
criterion based on the presence of particles < 25 nm in diameter and their consequent growth to Aitken mode. Event days
are days on which sub 25 nm particle formation and growth are observed. Non-event days are days on which neither
modes are present. Undefined days are the days which do not fit either criterion.

**2.2.2    Selecting non-cloudy days**

Cloudiness parameter ($P$) is the ratio of measured global radiation (Rd) divided by the theoretical global irradiance (Rg):

$$P = \frac{\text{Rd}}{\text{Rg}} (\text{Eq. 1})$$

The theoretical maximum of global radiation (Rg) is calculated by taking into consideration the latitude of the
measurement station and the seasonal solar cycle. While a complete cloud coverage is classified as $P < 0.3$, a clear-sky is
classified as $P > 0.7$ (Perez et al., 1990; Sogacheva et al., 2008; Sánchez et al., 2012). In Hyytiälä, the great majority of
NPF events are initiated during the morning hours after the sunrise, yet before the noon (Dada et al., 2017, in preparation).
Since the time of the sunrise varies widely in Hyytiälä between the different seasons, the time window 9:00-12:00 seems
a reasonable compromise for considering whether NPF did occur or not. We found that NPF events occurring outside our
selected time window were very few. Accordingly, in this work the days were classified as cloudy or clear-sky days based
on the median value of $P$ during 9:00-12:00 each day, corresponding to the time window for new particle formation.
Clear-sky days were those with a median of $P > 0.7$ between 9:00 and 12:00 and are the focus of this study. The median
value ensures that at least half of our selected time window is clear-sky while the rest can vary between clear-sky and
minor scattered clouds. The median is useful also because NPF is a regional-scale phenomenon, so for instance scattered
clouds on an otherwise sunny day affecting the local radiation measurements (and leading to a momentarily drop in $P$) do
not usually interrupt the regional NPF process. Clear-sky days were those with a median of $P > 0.7$ between 9:00 and
12:00 and are the focus of this study. For consistency, the variables compared in our study are taken from the same time
window, 9:00-12:00.

### 2.2.3    Sulfuric acid and oxidized organics proxies

The gaseous sulfuric acid concentration is estimated from a pseudo-steady-state-approximation proxy developed by Petäjä et al. (2009). This proxy takes into consideration the sulfuric acid source and sink terms as

$$[H_2SO_4]_{proxy} = k \cdot \frac{[SO_2] \cdot UVB}{CS} \text{ (Eq. 2).}$$

Here, UVB (W m$^{-2}$) is the fraction of the UV radiation reaching earth after being screened by ozone (280 – 320 nm) and the coefficient $k$ (m$^2$ W$^{-1}$ s$^{-1}$) is obtained from the comparison of the proxy concentration to the available measured H$_2$SO$_4$ data, and has a median value of 9.9×10$^{-7}$ m$^2$ W$^{-1}$ s$^{-1}$.

The concentration of monoterpene oxidation products, called oxidized organic compounds (OxOrg) here, is estimated using a proxy developed by Kontkanen et al. (2016b). This proxy is calculated by using the concentrations of different oxidants (the measured ozone concentration [O$_3$] and parameterizations for the hydroxyl and nitrate radical concentration, [OH] and [NO$_3$], respectively) and their reaction rates, $k_i$, with the monoterpenes. The MT $_{proxy}$ (in this case MT$_{proxy1,doy}$) is calculated by taking into account the effect of temperature-driven emissions, mixing of the boundary layer and the oxidation of monoterpenes, (Kontkanen et al., 2016b).

$$[OxOrg]_{proxy} = \frac{(k_{OH+MT}[OH] + k_{O3+MT}[O_3] + k_{NO3+MT}[NO_3]) \cdot MT_{proxy}}{CS} \text{ (Eq. 3).}$$

### 2.2.4    Particle formation rates

The formation rate of nucleation mode particles ($J_{3,C}$, particle diameter > 3 nm) was calculated based on the method suggested by Kerminen and Kulmala's equation (Kerminen and Kulmala, 2002). This quantity is a function of the calculated formation rate of 1.5 nm sized particles ($J_{1.5,C}$), their growth rate (GR) and the condensation sink (CS):

$$J_{3,C} = J_{1.5,C} \exp\left(-\gamma \frac{CS'}{GR_{1.5-3}}\left(\frac{1}{1.5} - \frac{1}{3}\right)\right), \text{ (Eq. 4)}$$

where $\gamma$ is a coefficient with an approximate value of 0.23 m$^3$ nm$^2$ s$^{-1}$. The value of $J_{1.5,C}$ was calculated by assuming heteromolecular nucleation between SA and OxOrg as follows:

$$J_{1.5,C} = K_{het}[H_2SO_4]_{proxy}[OxOrg]_{proxy}, \text{ (Eq. 5)}$$

The heterogeneous nucleation coefficient used in Eq. (5) is the median estimated coefficient for Hyytiälä scaled from Paasonen et al. (2010): $K_{het}$ = 9.2×10$^{-14}$ cm$^3$ s$^{-1}$. The scaling was made in order to fit the current data. The median value of [OxOrg] during the event days in April and May was found to be 1.6×10$^7$ cm$^{-3}$(Paasonen et al., 2010), whereas the revised median value of [OxOrg] by Kontkanen et al. (2016b) is 1.3×10$^8$ cm$^{-3}$. The scaling factor is the ratio between new and original [OxOrg] (0.1194). Accordingly, while the value of $K_{het}$ from Paasonen et al. (2010) is 1.1×10$^{-14}$ cm$^3$ s$^{-1}$, after the scaling by 0.1194 we obtain the revised $K_{het}$ = 9.2×10$^{-14}$ cm$^3$ s$^{-1}$.

The particle growth rate over the particle diameter range of 1.5−3 nm was calculated by taking into account the size of the condensing vapor molecule size and the thermal speed of the particle (Nieminen et al., 2010). The growth rates (1.5 – 3 nm) were calculated as 30-minute averages and as the sum of the growth rates due to the sulfuric acid (SA) vapor and OxOrg vapor condensation. The density of the particle was assumed to be constant (1440 kg/m$^3$). For SA, we first determined the SA concentration needed to make the particles grow at the rate of 1 nm/h by taking into account the mass of hydrated SA at the present RH and its density (Kurtén et al., 2007). Then, we calculated the GR of the particles due to SA condensation by using the SA proxy concentration. The same method was used for GR due to OxOrg condensation, where the vapor density was assumed to be 1200 kg/m$^3$ (Hallquist et al., 2009; Kannosto et al., 2008). Similarly, the GR due to OxOrg was calculated by using OxOrg proxy concentrations divided by the concentration needed for 1 nm/h GR.

### 2.2.5   Calculation of backward air-mass trajectories

Air mass trajectories were calculated using Hybrid Single-Particle Lagrangian Integrated Trajectory (HYSPLIT_4) Model at 96-hour backward trajectories at 100, 250 and 500 m arrival heights once per hour. Free access to transport model is developed and provided by NOAA (http://www.ready.noaa.gov/HYSPLIT.php). Input meteorological data required for the model were collected from GDAS (Global Data Assimilation System) archives.

## 3   Results and discussion

### 3.1  Effect of cloudiness on NPF

We studied NPF events as a function of cloudiness. Figure 1a shows the fraction of event, non-event and undefined days as a function of cloudiness parameter. We can see that clear-sky conditions favor the occurrence of NPF: the less clouds there were, the higher was the fraction of NPF event days. For instance, for days with the cloudiness parameter of 0.3 or less, the fraction of event days was less than 0.1 of the total classified days. However, the fraction of NPF event days reached a maximum of around 0.55 during complete clear-sky conditions ($P$ >0.7), with 877 days classified as NPF events, 560 undefined days and only 229 as non-events. On the NPF event days, the median cloudiness parameter $P$ during the time window 9:00-12:00 was found to be 0.75 (Fig. 1b), while the non-event days were characterized by lower values of $P$ (a median of around 0.25). Also, 75% of the NPF event days were found to have a cloudiness parameter larger than 0.5. The pattern found in Figure 1a follows from the fact that radiation seems essential for NPF at this site, as the events occur almost solely during daylight hours (Kulmala et al., 2004b). NPF is favored under abundant radiation conditions since the main components of freshly formed particles, are mainly formed photochemically (Petäjä et al., 2009; Ehn et al., 2014). The fraction of undefined days, however, remained constant regardless of cloudiness conditions.

Our results emphasize the fact that radiation favors NPF to occur, while clouds tend to decrease the probability of NPF. Undefined days were observed under cloudiness conditions that fell between those for NPF events and non-events. In general, undefined days can be interrupted NPF events, or unclassified plumes of small particles due to pollution (Buenrostro Mazon et al., 2009). The interruption of a NPF event can be due to a change in the measured air mass, or due to attenuation of solar radiation caused by the appearance of a cloud during the event. We will not consider undefined days further in our analyses.

The monthly variation of daily median cloudiness parameter within the time window of 9:00-12:00 during the classified
days is shown in Figure 2. Spring showed the best separation between the events and non-events in terms of the cloudiness
parameter, while the separation became weaker during the summer and specifically for June and July. Taken together,
Figures 1 and 2 emphasize the observation that the presence of clouds decreases the probability of NPF events.

**3.2  General character of NPF on clear-sky days**

Upon visualizing the cloudiness conditions during events and non-events, we chose a fixed constraint for clear-sky
conditions ($P > 0.7$) during the time window of NPF (9:00-12:00) and will next focus on other parameters that distinguish
NPF events from non-events.

The monthly distribution of the event fraction on clear-sky days appeared as double peaks in spring and autumn, with
spring having a higher fraction of events (Figure 3a). The minimum fraction of NPF events was recorded in December.
The fraction of non-event days peaked during winter with another peak in summer. The total number of NPF events varied
from year to year between 1996 and 2015. However, this variation did not show any specific trend of frequency (Figure
3b), which is in agreement with previous statistics reported from studies that did not consider clear-sky classification
(Nieminen et al., 2014).
**3.2.1    Backward air mass trajectories during clear-sky NPF events and non-events**

Since NPF is most frequent in spring, we dedicated our focus into this season (Figure 3a). The springtime medians and
percentiles of air-mass trajectories arriving at Hyytiälä during clear-sky NPF events and non-events were calculated 96
hours backward in time at the 100-m, 250-m and 500-m arrival heights for the years 1996-2015. The medians and similarly
the percentiles were calculated by taking the median compass direction at every point on the trajectory (1 hour between
every two points), arriving every half an hour at Hyytiälä. The trajectories arriving at Hyytiälä at these three heights were
quite similar, and those arriving at the 500-m height are shown in Figure 4. Medians and percentiles of the routes were
calculated by taking the median of the trajectories at every half hour for spring time NPF event days and non-event days
separately. During the NPF event days, the measured air masses were found to originate mainly from the north and to
pass over Scandinavia before arriving at Hyytiälä. Similar to previously reported results, air masses arriving from the
north and north-west directions result in clean air with low pollutant (particulate matter and trace gas) concentrations
(Nieminen et al., 2015). During NPF the non-event days, air masses originated from more polluted areas in Europe and
Russia, resulting in elevated levels of condensation sink and other air pollutants in Hyytiälä, as also seen in previous
studies (Sogacheva et al., 2005).
**3.2.2    Influences of CS, meteorological parameters and trace gases**

In Figure 5a we present the monthly variation of condensation sink during NPF events and non-events under daytime
clear-sky conditions. NPF events tended to be favored by low values of CS throughout the year. In all the months except
during summer, the 75th percentile of the event day values of CS was lower than the 25th percentile of the non-event day
values of CS. On the NPF event days, CS had its maximum in summer, which might be one of the main reasons for the

local minimum in the NPF event frequency during the summer months (Figure 3a). However, the monthly cycle of CS during non-event days had two maxima, one in spring and another one in autumn, which might suggest that during these seasons, high values of CS prevented NPF to occur on particular days. The difference in the value of CS between the NPF event and non-event days was the highest in March and the lowest during the summer months.

Figure 5b shows the monthly temperature conditions ($T$) during the daytime NPF events and non-events. While higher temperatures favored NPF during the months when the average temperature was below 273.15 K ($0^o$ C) (months 1, 2, 3, 11 and 12), the opposite was true at average temperatures above 273.15 K ($0^o$ C). The highest recorded temperature at which an event occurred during $P > 0.7$ sky was 300 K (25 $^o$C) and the minimum temperature was 252 K (-21 $^o$C). Accordingly, both very high and very low temperatures were not favorable conditions for NPF to occur. Although an increase in the ambient temperature results in higher concentrations of monoterpenes due to increased emissions, thereby favoring new particle formation and growth (Kulmala et al., 2004), Figure 5b shows that very high temperatures tend to suppress NPF. This latter feature is at least partly related to the positive relation between the ambient temperature and pre-existing aerosol loading (and hence CS) in Hyytiälä (Liao et al., 2014), even though it might also be attributed to the increase in vapor evaporation coefficients, which results in less stable clusters at high temperatures (Paasonen et al., 2012).

Consistent with an earlier study (Hamed et al., 2011), our results indicate that NPF is favored by low values of ambient relative humidity in Hyytiälä (Fig. 5c). This observation does not conflict with chamber experiments (e.g.Duplissy and Flagan, 2016) or theory (Merikanto et al., 2016; Vehkamäki et al., 2002), which suggest higher nucleation rates at higher values of RH, because binary $H_2SO_4$–water nucleation is not expected take place in Hyytiälä. Other studies have proposed that increased RH limits some VOC (Volatile Organic Compounds) ozonolysis reactions, preventing the formation of come condensable vapors necessary for nucleation (Boy and Kulmala, 2002). This might partially explain the observed anti-correlation between RH and particle formation rates. Therefore, it seems plausible that RH affects NPF via atmospheric chemistry rather than via changing the sink term for condensing vapors and small clusters. Additionally, we found clear differences in how trace gas concentrations were associated with RH between the NPF event and non-event days (Table 1). For instance, $O_3$ showed a strong negative correlation with RH during events and non-events. However, during non-event days, a positive correlation appears between RH and each of CO, $SO_2$ and NOx while the correlation between those seems to be absent during event days. Our results show that air masses coming from central Europe and passing over the Baltic Sea tend to have higher values of RH.

After looking at the characteristics of clear-sky NPF event and non-event days in terms of meteorological parameters and CS, we looked at the variation of trace gas (CO, $SO_2$, NOx and $O_3$) concentrations during these conditions (Figure 6). Out of these gases, at least $SO_2$ and $O_3$ are expected to enhance NPF, $SO_2$ as a precursor for sulfuric acid and $O_3$ as an oxidant forming ELVOCs (Extremely Low Volatile Organic Compounds) (Donahue et al., 2012; Ehn et al., 2014). However, none of these vapors seemed to have higher concentrations during NPF event days. This suggests that, as tracers of pollution, these gases are strongly linked with high anthropogenic CS, so air masses having high trace gas concentrations often do not result in NPF in Hyytiälä.

### 3.3 Connection of nucleating precursor vapors with new particle formation rate

#### 3.3.1 Precursor vapor proxies

In this study, we determined $J_{1.5,C}$ using the proxies for both SA and OxOrg. The monthly variations of these precursors (in the time window 9:00-12:00) are shown in the Figure 7. During clear-sky conditions, the SA proxy tended to have the highest median daytime values during the winter months with a maximum in February (Figure 7a). Contrary to this, the seasonal distribution of the SA proxy reported in Hyytiälä appears as double peaks with an absolute maximum in spring and a smaller one in autumn when presenting the data without excluding cloudy days (Nieminen et al., 2014). During winter, both condensation sink and boundary layer height are lower than in the summer (Paasonen et al., 2013), which might explain the higher concentrations of SA during the winter months.

Being a function of temperature, the OxOrg proxy concentration was generally found to follow the monthly cycle of the ambient temperature. The median value of [OxOrg] was higher on NPF events days during every month compared with non-event days (Figure 7b). The largest difference in [OxOrg] between the NPF events and non-events, in terms of its median value, was recorded for January and the least difference for May. It is to be noted that the proxy values represent the measured values less accurately during the winter than during the other periods (Kontkanen et al., 2016b).

#### 3.3.2 Particle formation rates

The calculated new particle formation rate, $J_{1.5,C}$, approximated with Eq. (5) shows a similar behavior as the [OxOrg] (see Figures 7 and 8), being higher for the clear-sky NPF event days in comparison with non-event days. Also, the difference in the value of $J_{1.5,C}$ between the NPF events and non-events was the highest in the winter, and the lowest in summer. The monthly cycle of $J_{1.5,C}$ followed closely that of [OxOrg], as the latter had a higher seasonal variability than the sulfuric acid proxy concentration, being thereby capable of affecting the seasonal pattern of $J_{1.5,C}$ (Figure 8a). The diurnal cycle of $J_{1.5,C}$ during the NPF event days showed an increase along with sunrise, a peak at midday and decrease along with sunset. However, for non-event days the $J_{1.5,C}$ value was relatively constant throughout the day and had clearly lower values than during the NPF event days (Figure 8b).

Since previous studies have shown that there is a clear difference in observed $J_3$ between the event and non-event days, and much less difference in observed $J_{1.5}$ (Kulmala et al. 2013), we decided to focus on $J_3$ in our event to non-event discrimination. Previous studies which did not consider clear-sky conditions have reported values of observed spring time $J_3$ between 0.01 and 5 cm$^{-3}$ s$^{-1}$ (median = 0.94 cm$^{-3}$ s$^{-1}$ ) during the period of active NPF (Kulmala et al. 2013). Our values of $J_{3,C}$ fit between the extremes of these values for the spring time and time window 9:00 to 12:00, with a slightly higher median value of 1.9 cm$^{-3}$ s$^{-1}$ (Figures 9 a,b). The formation rate of 3 nm particles is affected not only by the new particle formation rate ($J_{1.5}$) but also by the scavenging of newly-formed particles by coagulation into pre-existing particles. We found that, in general, the values of $J_{3,C}$ calculated using Eq. (4) and (5) were higher on NPF event days compared with non-event days in all months (Figure 9a). The difference between the event and non-event days was the largest in winter and decreased towards summer. However, the diurnal cycles of percentiles and medians of $J_{3,C}$ during each month peaked around noon for both NPF events and non-events. One example is presented in Figure 9b, showing that $J_{3,C}$ tended to

increase after the sunrise, to peak at about midday and to diminish after sunset. This kind of diurnal cycle was similar for
all the months. Hourly values of $J_{3,C}$ calculated during the NPF event days were higher than those during the non-event
days. During the spring months, the difference in the median $J_{3,C}$ between the NPF events and non-events, calculated for
every half an hour, appeared to increase at about 10:00 and then it started to decrease again at about 13:00 (Figure 9b).
On NPF event days, in comparison to springtime $J_{1.5,C}$ which peaked at around 10:45 (Figure 8b), $J_{3,C}$ peaked typically
about half-an hour later. This time delay indicates how long it takes for the particles grow from 1.5 nm to 3 nm. This
growth is a critical step of NPF (Kulmala et al. 2013), and depends on concentrations of available vapor precursors.

In Figure 10 we present the median diurnal cycles of $J_{3,C}$ plotted against the median diurnal cycles of CS during classified
clear-sky NPF events and non-events. The diurnal cycle was calculated by taking the median CS at every half hour
throughout the season. On the NPF event days, the CS had higher values during the nighttime and lower values during
daytime with a minimum at noon. It is important to remember that J was calculated only for daytime when the SA proxy
was available (UV-B radiation is needed for the proxy). On non-event days, the values of CS showed no clear diurnal
pattern, had practically no difference between the daytime and nighttime hours, and were roughly twice those recorded
during the clear-sky NPF event days. The difference in CS between NPF events and non-events follows from the distinctly
different air masses arriving at Hyytiälä. For instance, it has been shown that air-masses originating from the north and
passing over Scandinavia have, on average, lower values of CS than the air masses passing over Russia and central Europe
(Sogacheva et al., 2005; Nieminen et al., 2015).

On NPF event days, the median approximated formation rate of 3 nm particles had its maximum value at about midday
and was significantly higher than on non-events days (Figures 9b and 10). A clear negative relation could be seen between
the median seasonal diurnal cycles of CS and $J_{3,C}$ on NPF event days (specifically during spring daytime) (Figure 10).
This kind of relation was not observed during non-event days when these two quantities seemed to be independent of
each other (Figure 10). In summer, the median value of $J_{3,C}$ was roughly similar between NPF events and non-events,
whereas the median value of CS was almost ten times higher during the non-event days compared with event days. The
high values of $J_{3,C}$ for the non-event days in summer, despite the high CS values, seem to suggest that some other factor
limits the actual NPF rate. One possibility is that freshly-formed clusters are rapidly evaporated due to higher ambient
temperatures (see Fig. 5b). This will be discussed in a more detail in the following section. Higher values of CS on non-
event days are expected, bearing in mind that these particles act as surfaces for scavenging precursor gases and freshly
formed particles (Hussein et al., 2008). The association of a high CS with the lower NPF probability has been observed
in many studies conducted in Hyytiälä (Boy and Kulmala, 2002; Hyvönen et al., 2005; Baranizadeh et al., 2014), as well
as in other rural and urban areas, including Egbert and Toronto in Canada (Jun et al., 2014), Preila in Lithuania (Mordas
et al., 2016), Po Valley in Italy (Hamed et al., 2007) and Budapest and K-puszta in Hungary (Salma et al., 2016).

**3.3.3    Threshold separating the NPF events and non-events**

Since quite a visible separation could be observed in the calculated values of $J_{3,C}$ between the spring-time clear-sky NPF
events and non-events, and since $J_{3,C}$ had its maximum at around midday, the plot of CS versus temperature at midday
(11:00-12:00) in spring provides an equation that effectively separates the NPF events from non-events during this season
(Figure 11). This equation was determined using a linear discriminant analysis (LDA) similar to Hyvönen et al. (2005).
The equation provides a line that separates NPF events from non-events at 95% confidence towards non-events. Based
on their midday CS and Temperature, the data point follows either classes. More specifically, the days with
$CS$ (s$^{-1}$) > -3.091×10$^{-5}$ × $T$ (in Kelvin) + 0.0120, (Eq. 6)
lie above the threshold line. Almost no non-event days fall below this line (< 5%). The points above the line were also
characterized with higher trace gases concentrations and lower calculated formation rates of 3 nm particles than the rest
of the points.
The separation between the clear-sky NPF events and non-events in the CS versus $T$ plot was less evident in autumn and
disappeared completely in the summer and winter (Figure 12). Interestingly, a large number of NPF event days during
these seasons still fell below the threshold line given by Eq. (6). Furthermore, we analyzed the effect of RH in separating
the events from nonevents, similar to the study done on RH by Hyvönen et al. (2005). We found that compared with CS
vs temperature data, depicting CS vs RH (data not presented) did not work better in separating NPF events from non-
events during clear-sky conditions.
### 3.3.4 Probability of NPF events and non-events
Since the biggest difference in the calculated 3 nm particle formation rates between the NPF events and non-events was
observed around noon (Figure 9b), and since CS and temperature showed promising threshold values for predicting the
occurrence of NPF non-events during spring (up to 95%) (Figure 11), Figure 13 presents the probability of having a NPF
event in Hyytiälä at a specific CS and temperature within the time window 11:00-12:00. The probability was calculated
taking the fraction of events to the total events and non-events in every cell which is 2.5 K on the x-axis and a ratio of
1.14 on the y-axis between every two consecutive CS values. The highest probability of having a NPF event corresponded
to conditions having moderate temperatures and low values of CS. At high values of CS, there was a zero probability for
NPF regardless of the temperature. However, at moderate and low values of CS, the probability of having a NPF event
decreases as we go to lower temperature. This could be explained by lower emissions of VOCs, and thus lower OxOrg
concentrations, at lower temperatures. Similarly, the probability of NPF decreases as we go to higher temperatures at
constant values of CS. This latter feature might be attributed to conditions unfavorable for clustering due to high
temperatures. Although previous studies have developed criteria for NPF probability which could work in diverse
environments(Kuang et al., 2010), they did not explore the dependency of their parameter on atmospheric conditions.
### 4 Conclusion
In this study we combined 20 years of data collected at the SMEAR II station in order to characterize the conditions
affecting the frequency of NPF events in that location. By focusing only on clear-sky conditions, we were able to get new
insight into differences between the NPF events and non-events. In clear-sky conditions, the meteorological conditions,
trace gas concentrations and other studied variables on NPF event days appeared to be similar to those presented in the
previous studies which did not consider clear-sky classification. Furthermore, the monthly data refined the analysis so
that the differences caused by different quantities became more visible compared the previous studies conducted for this
site. Our work confirms, with a complementary dataset, the conclusions of Baranizadeh et al. (2014) that NPF events and
non-events are typically associated with clear-sky and cloudy conditions, respectively.

Our results showed that using SA and OxOrg proxies to calculate the apparent formation rates of 1.5 and 3 nm particles
works well in differentiating the clear-sky NPF events from non-events. Moreover, during clear-sky conditions the effect
of CS on attenuating or even preventing NPF was quite visible: CS was, on average, two times higher on the non-event
days compared with the NPF event days. Similarly, many other meteorological variables affected NPF. By using CS and
ambient temperature, we were able to find a threshold above which no clear-sky NPF events occurred. This threshold is
described with an equation that is able to separate 97.4% of the NPF events from non-events during spring time. In clear
sky conditions, when there is plenty of radiation available, NPF events take place as long as the CS is low enough and
temperature is moderate. Although a weaker separation was observed in the other seasons, considering only clear-sky
conditions enabled us to form a map of the probability of having a NPF event within specific CS and temperature
conditions. Using clear-sky conditions appears to bring us one step forward towards understanding NPF and predicting
their occurrences in Hyytiälä. Our study serves as a basis to future detailed comparisons with observations to formulate
even more robust conclusions.

**5   Acknowledgements**

This work was supported by the Academy of Finland Centre of Excellence program (grant no. 272041) and Nordic Top-
level Research Initiative (TRI) Cryosphere-Atmosphere Interactions in a Changing Arctic Climate (CRAICC). The
authors thank the division of atmospheric sciences at the University of Helsinki. We also thank Mrs. Ksenia Tabakova
for providing air-mass trajectory data.

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

**Table 1** Correlation coefficients between different meteorological parameters, gas concentrations and condensation sink (CS) during clear-sky events and non-events during spring (Mar-May, 1996-2015) and time window 9:00 – 12:00. The light blue refers to medium positive correlation (>0.45) and the dark blue to high positive correlation (>0.7), light orange refers to medium negative correlation (<-0.45) and dark red to high negative ones (<-0.7).

| | CS | T | RH | CO | NO$_X$ | SO$_2$ | O$_3$ |
|---|---|---|---|---|---|---|---|
| **Events** | | | | | | | |
| **CS** | 1 | | | | | | |
| **T** | 0.28 | 1 | | | | | |
| **RH** | -0.06 | **-0.64** | 1 | | | | |
| **CO** | 0.33 | -0.37 | 0.26 | 1 | | | |
| **NO$_X$** | 0.53 | -0.19 | 0.21 | 0.47 | 1 | | |
| **SO$_2$** | 0.4 | -0.29 | 0.14 | 0.36 | 0.58 | 1 | |
| **O$_3$** | 0.23 | 0.52 | **-0.51** | -0.06 | -0.08 | -0.08 | 1 |
| **Non-Events** | | | | | | | |
| **CS** | 1 | | | | | | |
| **T** | 0.15 | 1 | | | | | |
| **RH** | -0.12 | **-0.81** | 1 | | | | |
| **CO** | 0.53 | **-0.68** | 0.5 | 1 | | | |
| **NO$_X$** | 0.34 | **-0.51** | 0.45 | **0.7** | 1 | | |
| **SO$_2$** | 0.23 | **-0.55** | 0.42 | 0.56 | 0.41 | 1 | |
| **O$_3$** | 0.43 | 0.62 | **-0.64** | -4E-04 | -0.07 | -0.13 | 1 |

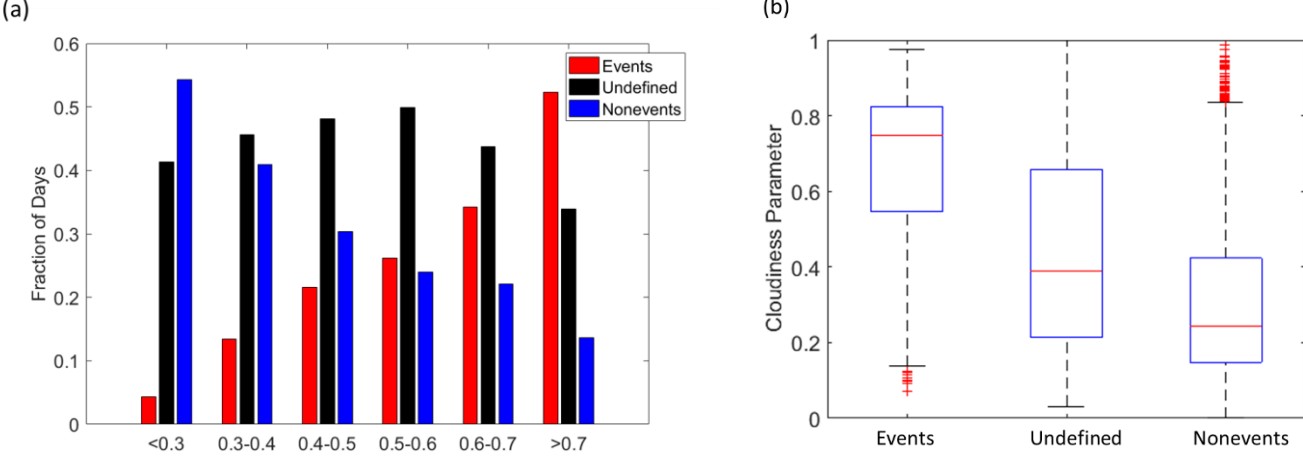

**Figure 1: (a)** Figure showing the fraction of days which are classified as NPF events, non-events, and undefined days during different sky cloudiness conditions. **(b)** Cloudiness daily (9:00 – 12:00) medians and percentiles recorded during NPF event, undefined and non-event days. The red line represents the median of the data and the lower and upper edges of the box represent 25th and 75$^{th}$ percentiles of the data respectively. The length of the whiskers represent 1.5 x interquartile range which includes 99.3% of the data. Data outside the whiskers are considered outliers and are marked with red crosses.

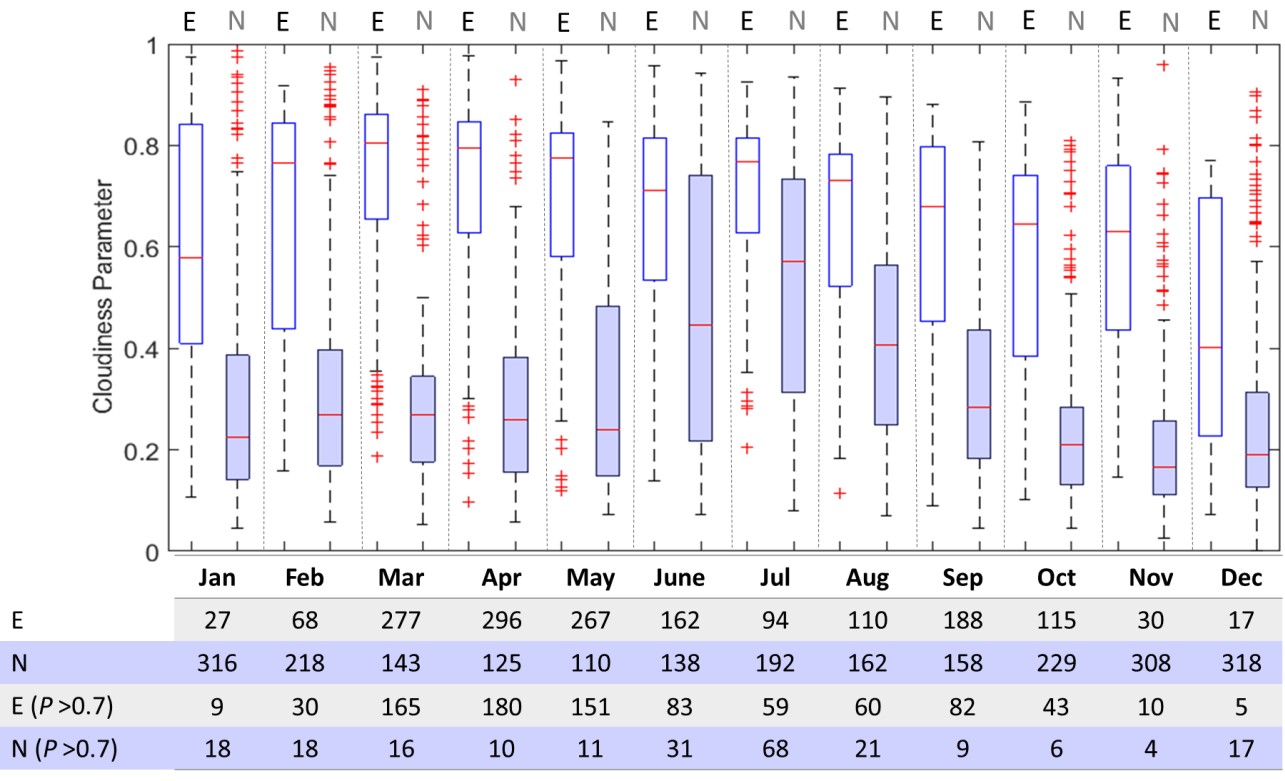

| | Jan | Feb | Mar | Apr | May | June | Jul | Aug | Sep | Oct | Nov | Dec |
|---|---|---|---|---|---|---|---|---|---|---|---|---|
| E | 27 | 68 | 277 | 296 | 267 | 162 | 94 | 110 | 188 | 115 | 30 | 17 |
| N | 316 | 218 | 143 | 125 | 110 | 138 | 192 | 162 | 158 | 229 | 308 | 318 |
| E (*P* >0.7) | 9 | 30 | 165 | 180 | 151 | 83 | 59 | 60 | 82 | 43 | 10 | 5 |
| N (*P* >0.7) | 18 | 18 | 16 | 10 | 11 | 31 | 68 | 21 | 9 | 6 | 4 | 17 |

**Figure 2: Monthly variation of cloudiness daily (9:00 – 12:00) medians and percentiles recorded during NPF events (E; white)**
**and non-events (N; shaded). Numbers below the plot correspond to the number of data points included in each boxplot. Number**
**of clear-sky events (E (P>0.7)) and clear-sky non-events (N (P>0.7)) accompany the plot. See Figure 1 for explanation of**
**symbols.**


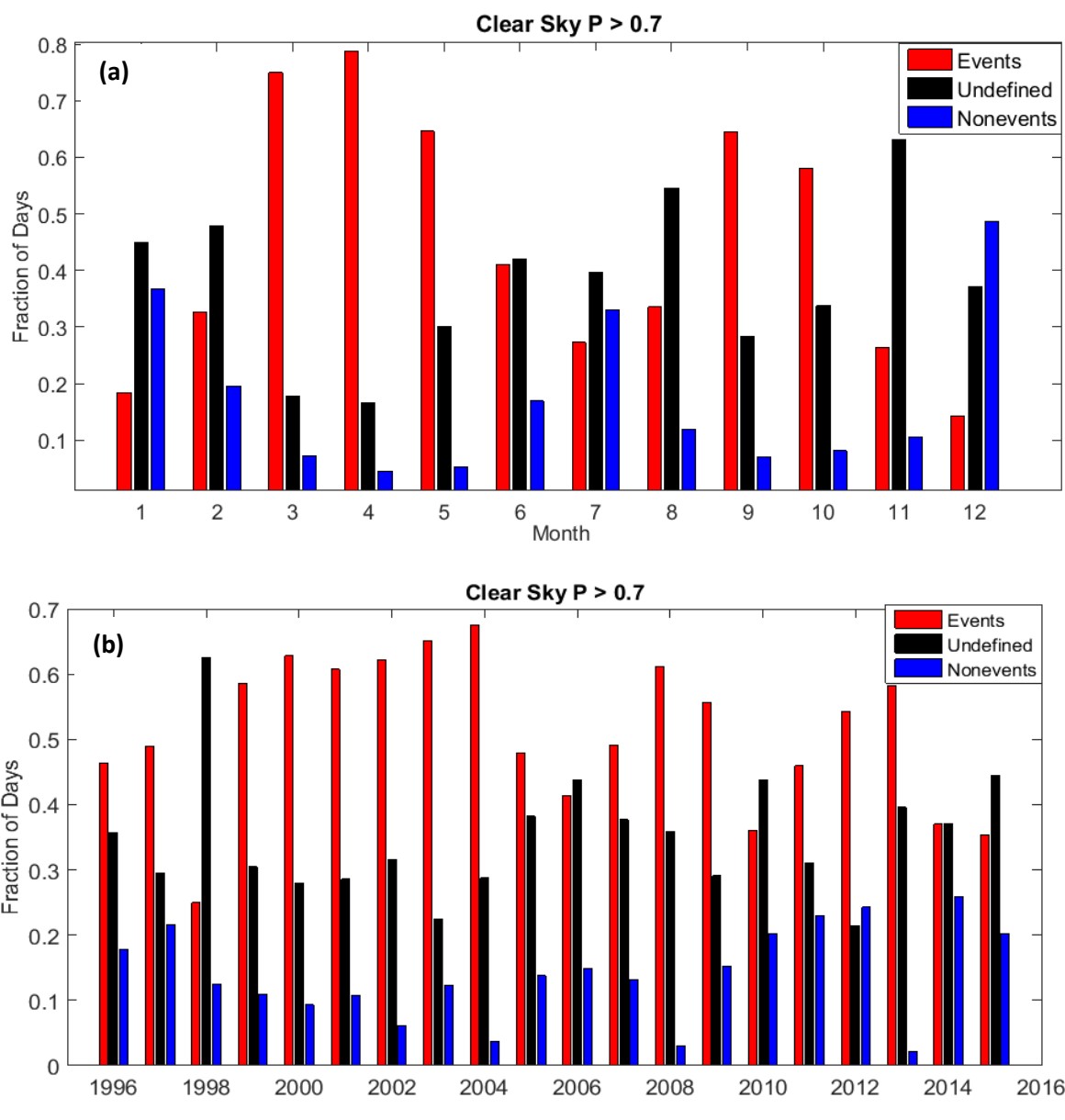

**Figure 3: (a) Monthly and (b) yearly fraction of clear-sky days classified as NPF Events, undefined and non-events. In year 1998, global radiation data is limited to 5.4%, making the classification bias.**


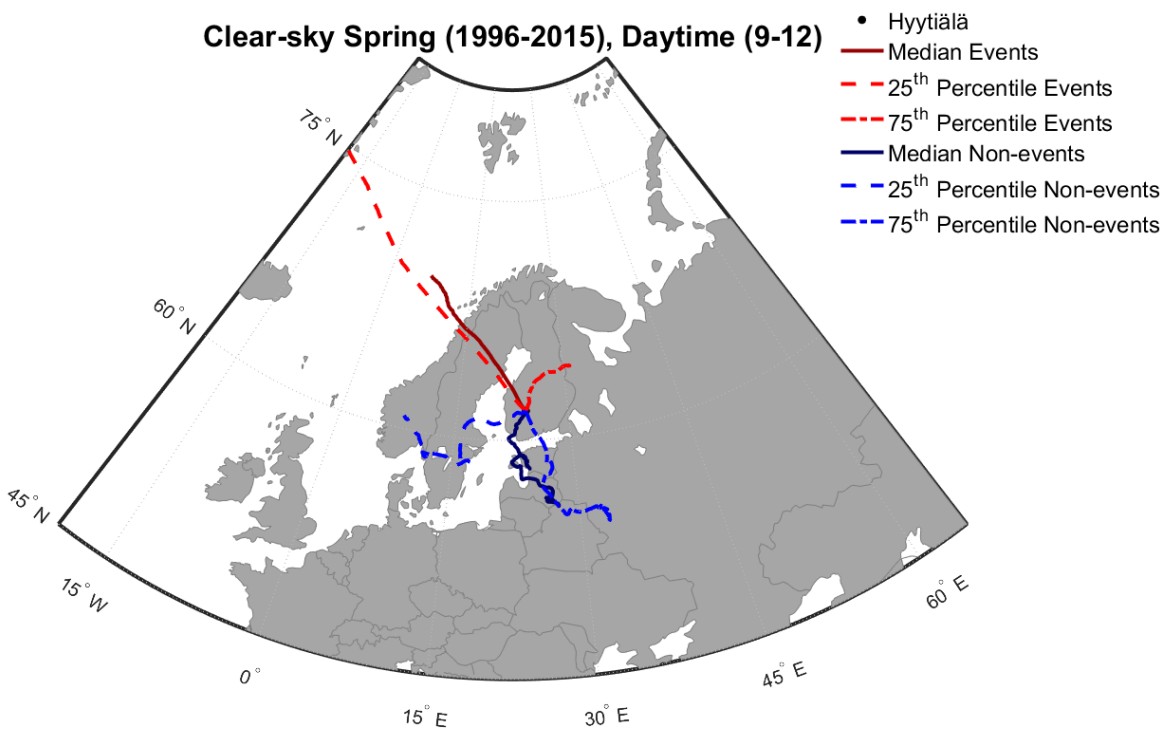


**Figure 4: Median and percentiles of 96 hours backward air-mass trajectories arriving to Hyytiälä during spring time (9:00-**
**12:00).**

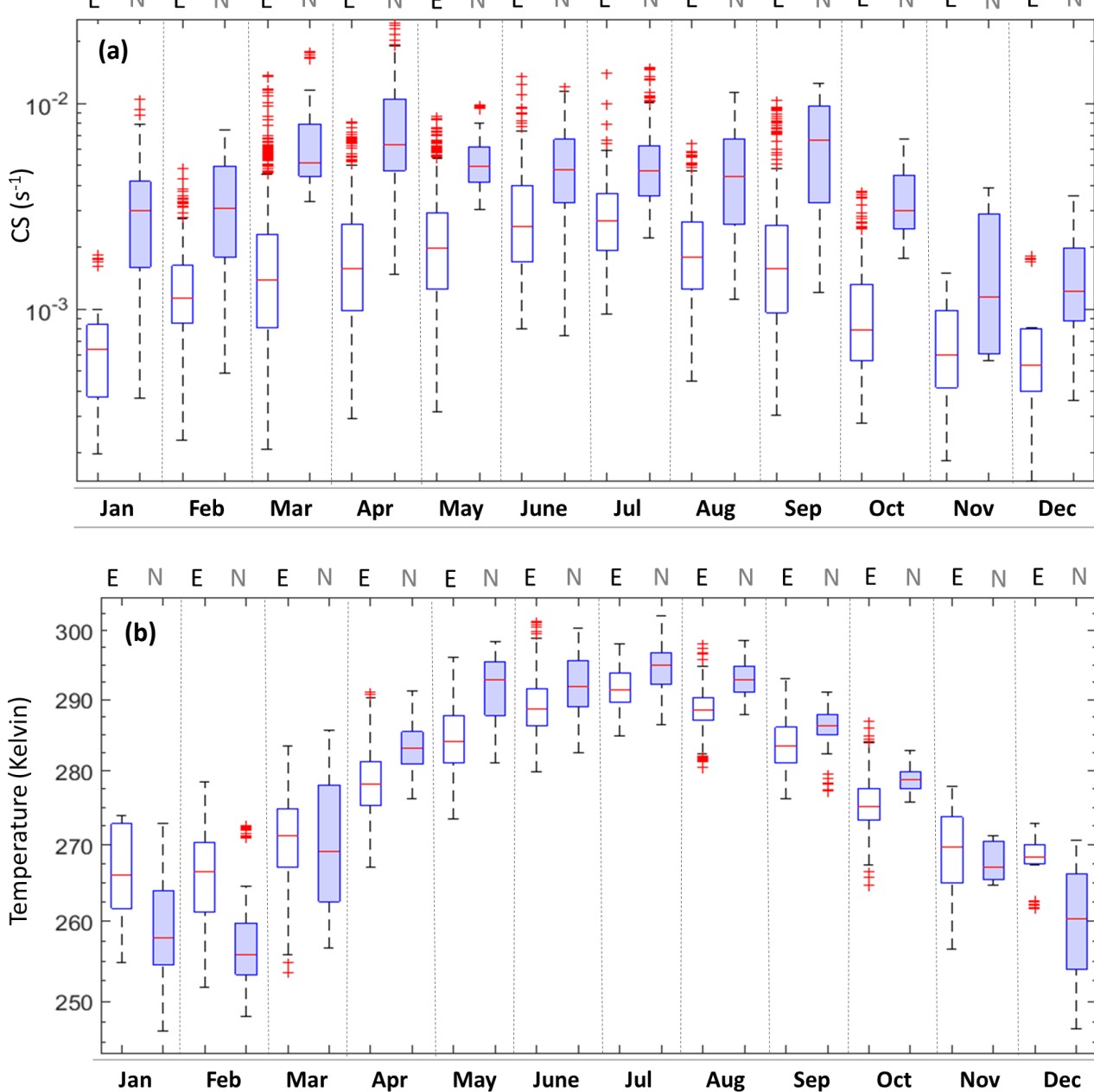

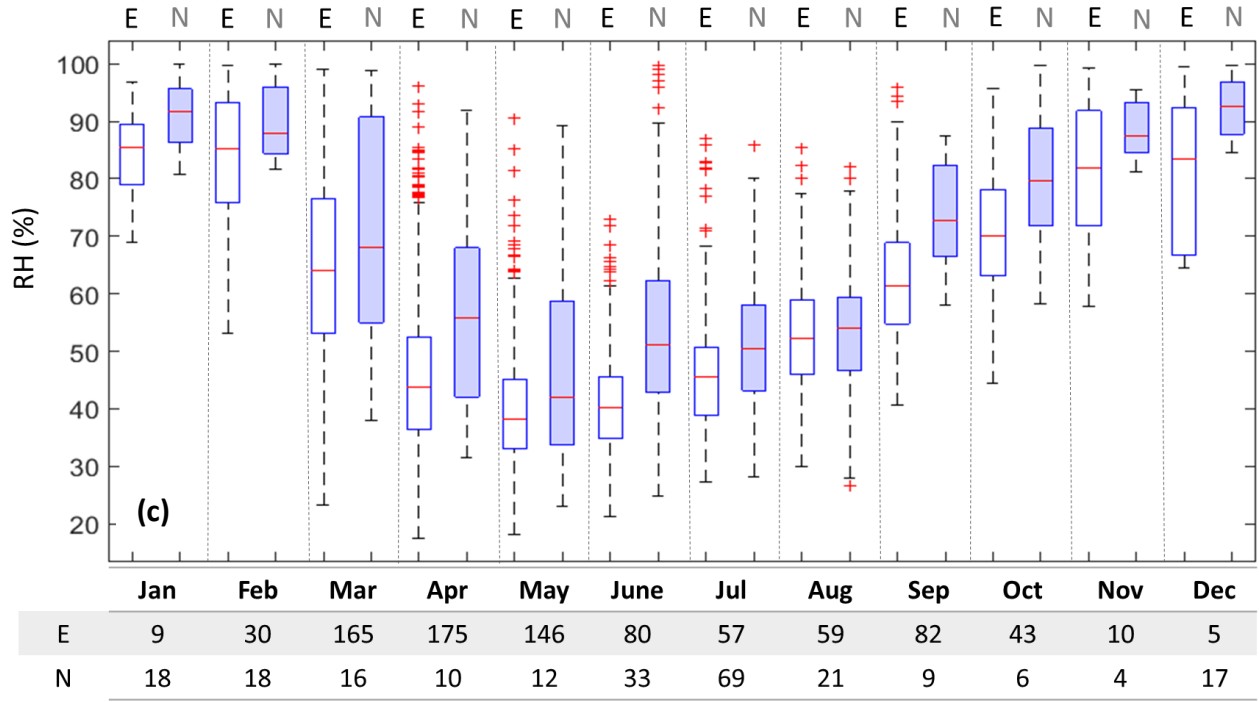

| | **Jan** | **Feb** | **Mar** | **Apr** | **May** | **June** | **Jul** | **Aug** | **Sep** | **Oct** | **Nov** | **Dec** |
|---|---|---|---|---|---|---|---|---|---|---|---|---|
| E | 9 | 30 | 165 | 175 | 146 | 80 | 57 | 59 | 82 | 43 | 10 | 5 |
| N | 18 | 18 | 16 | 10 | 12 | 33 | 69 | 21 | 9 | 6 | 4 | 17 |

**Figure 5: Median and percentiles of monthly variation (9:00 – 12:00) at P>0.7 of (a) CS (b) Temperature and (c) RH during NPF events (E, white) and non-events (N, shaded). See Figure 1 for explanation of symbols.**


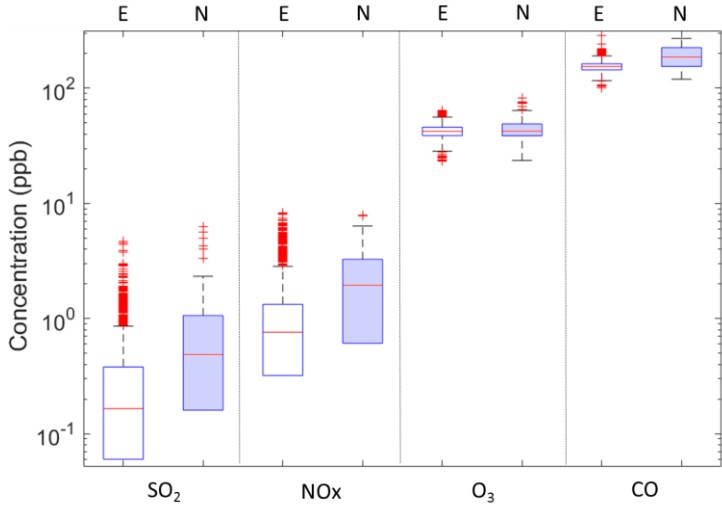

**Figure 6: Spring time (months 3,4,5) medians and percentiles of trace gases during clear-sky events (E, white) and non-events (n, shaded) during daytime (9:00 – 12:00). See Figure 1 for explanation of symbols.**



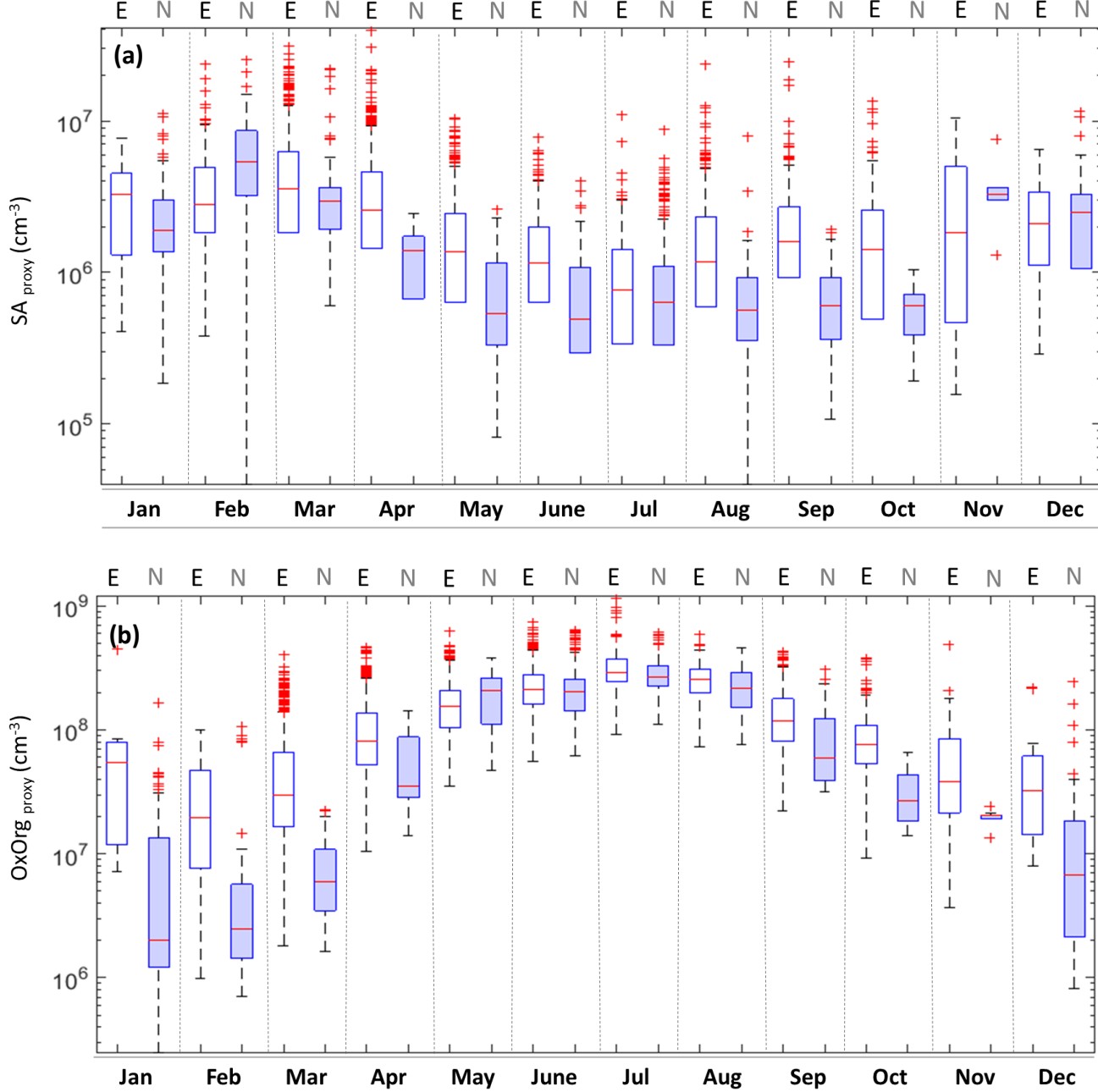

**Figure 7: Monthly variation of medians and percentiles of (a) SA proxy and (b) OxOrg proxy at P>0.7 during the time window 9:00 – 12:00 of NPF events (E, white) and non-events (N, shaded). See Figure 1 for explanation of symbols.**


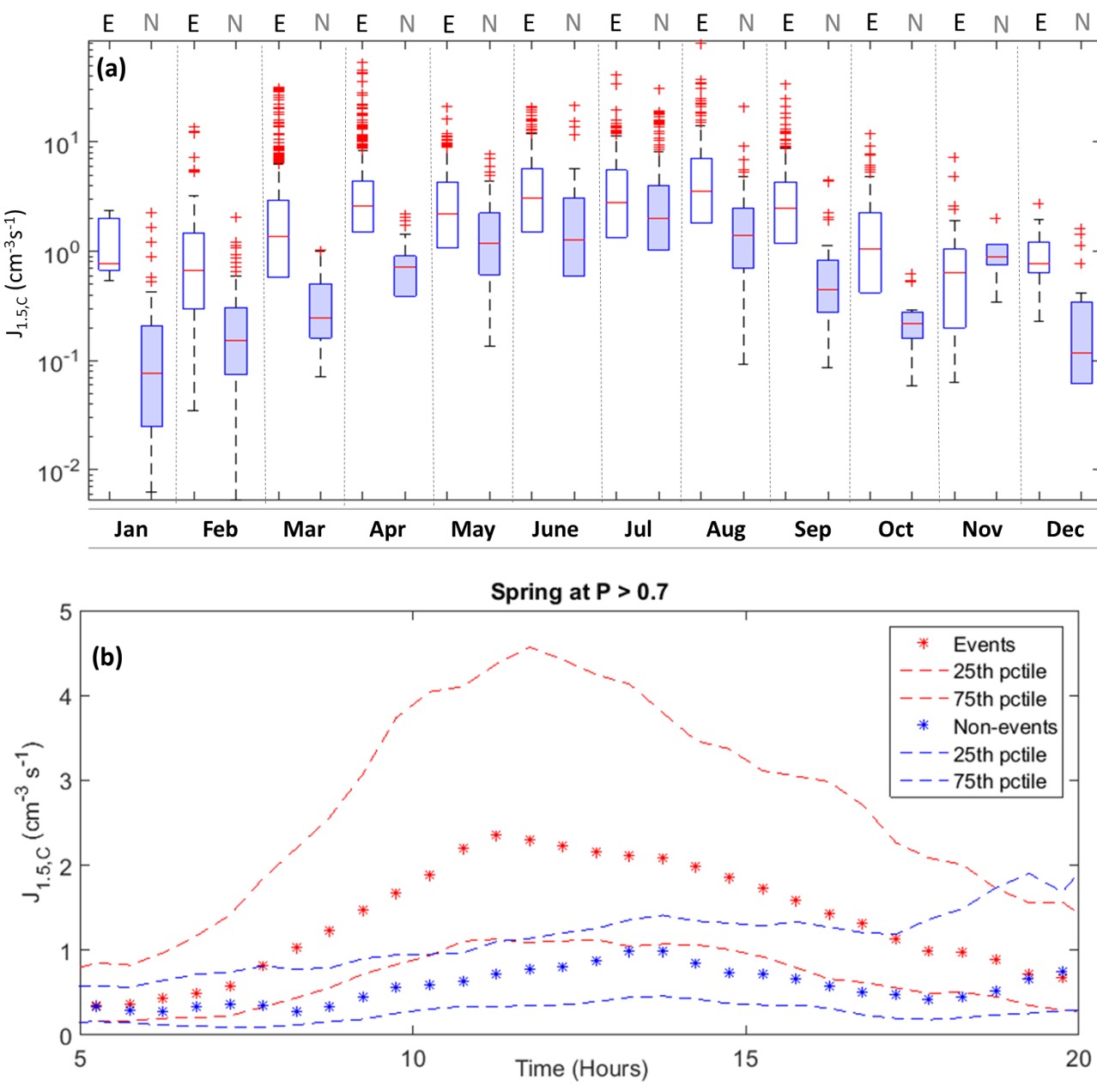

**Figure 8: (a) Monthly variation of medians and percentiles of $J_{1.5,C}$ during the time window 9:00 – 12:00 of NPF events (E, white) and non-events (N, shaded). See Figure 1 for explanation of symbols. (b) The diurnal cycle of $J_{1.5,C}$ during Spring. The nighttime is missing in this plot due to unavailable SA proxy which uses UVB to be calculated.**



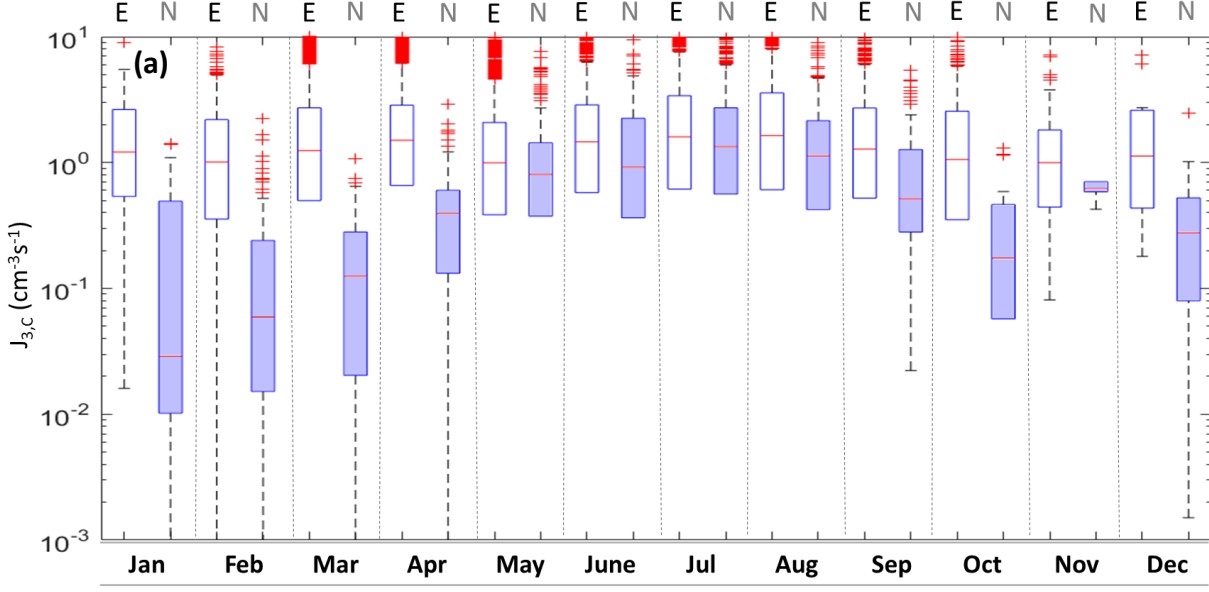


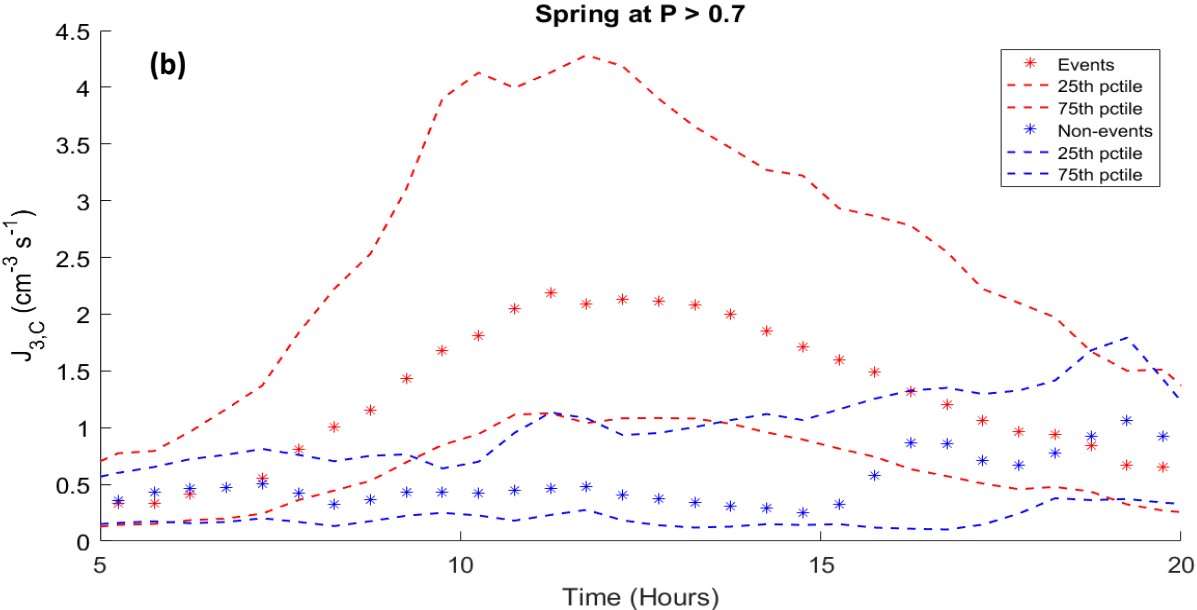

**Figure 9: (a)** $J_3$,c medians and percentiles during different months separated classified NPF events (E, white) and non-events
(N, shaded) (9:00 -12:00). See Figure 1 for explanation of symbols. **(b)** The diurnal cycle of $J_{3,C}$ during spring. The nighttime is
missing in this plot due to unavailable SA proxy which uses UVB to be calculated.


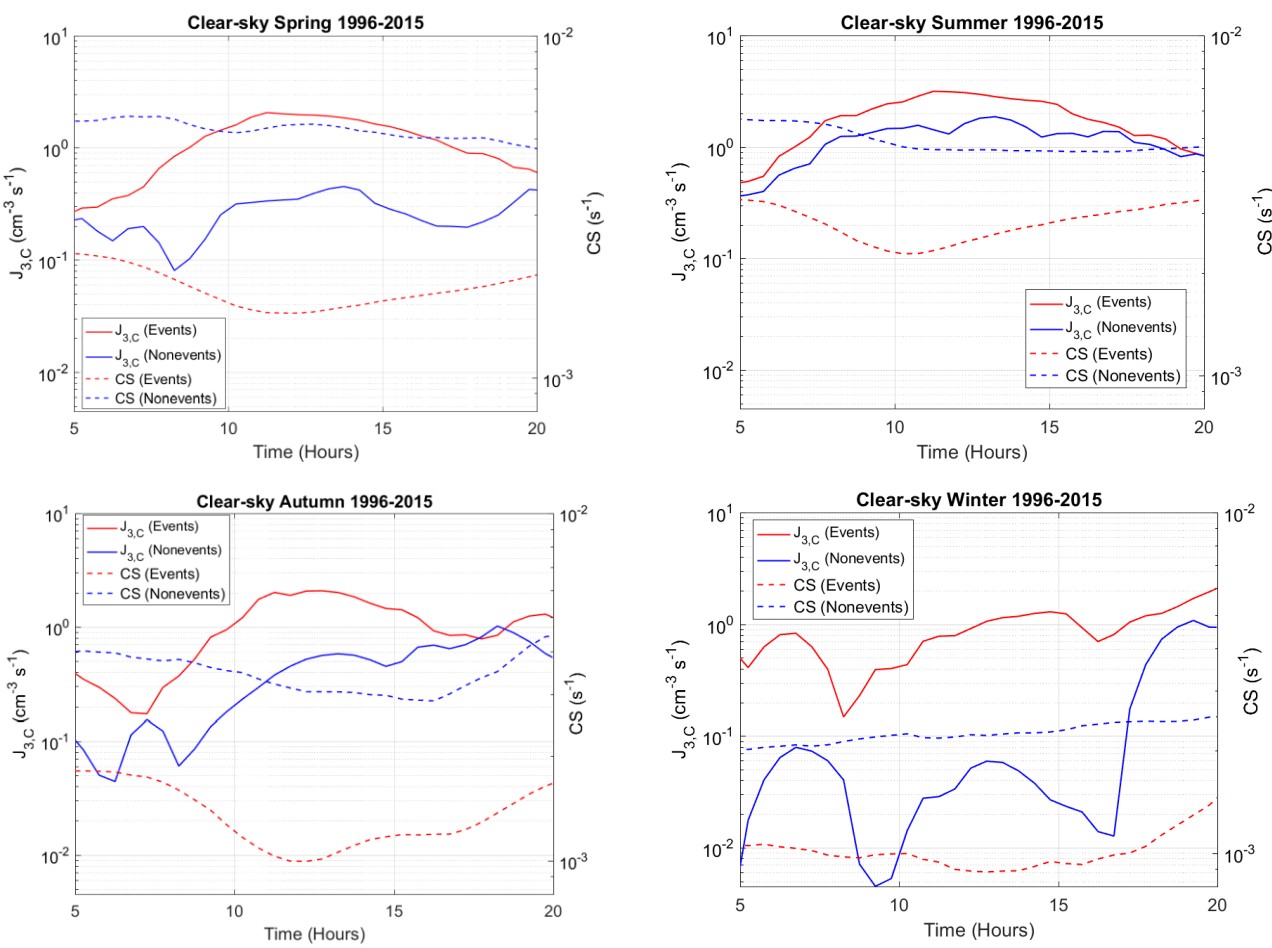


Figure 10: Diurnal cycle of median values of calculated formation rate of 3 nm particles ($J_{3,C}$) and condensation sink (CS) during different seasons on clear-sky events and non-events.


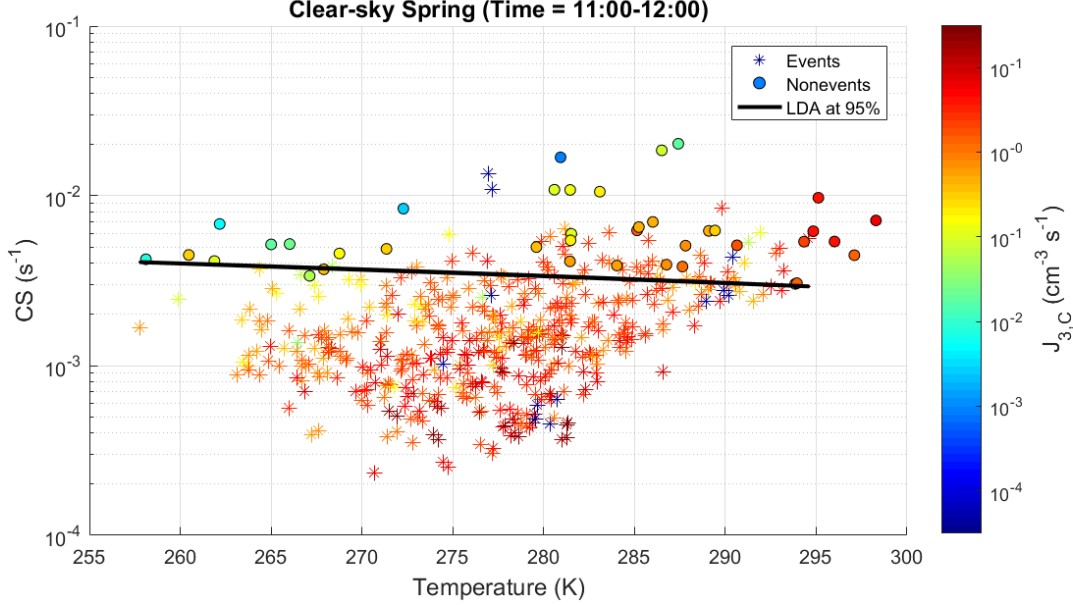


**Figure 11: Relationship between temperature and CS during spring time (11:00 – 12:00) NPF clear-sky (P > 0.7) event days**
**and non-event days color-coded with $J_{3,C}$. Horizontal line is calculated from LDA at 95% confidence relative to nonevents and**
**is demonstrated by Eq. (6).**

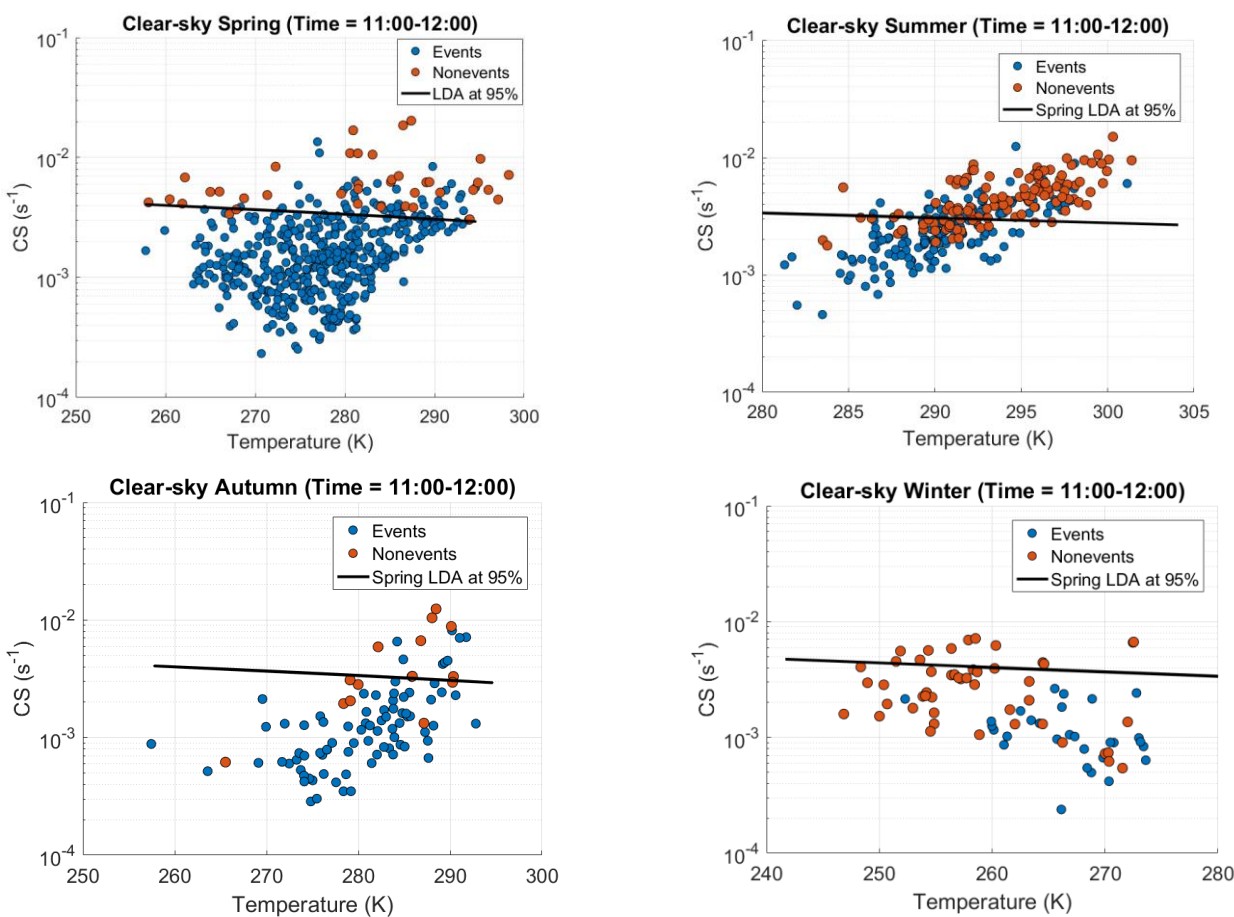

**Figure 12: Relationship between CS and Temperature (time window: 11:00 – 12:00) NPF clear-sky event days and non-event days. Horizontal line is calculated from spring LDA at 95% confidence relative to non-events and is demonstrated by Eq. (6).**

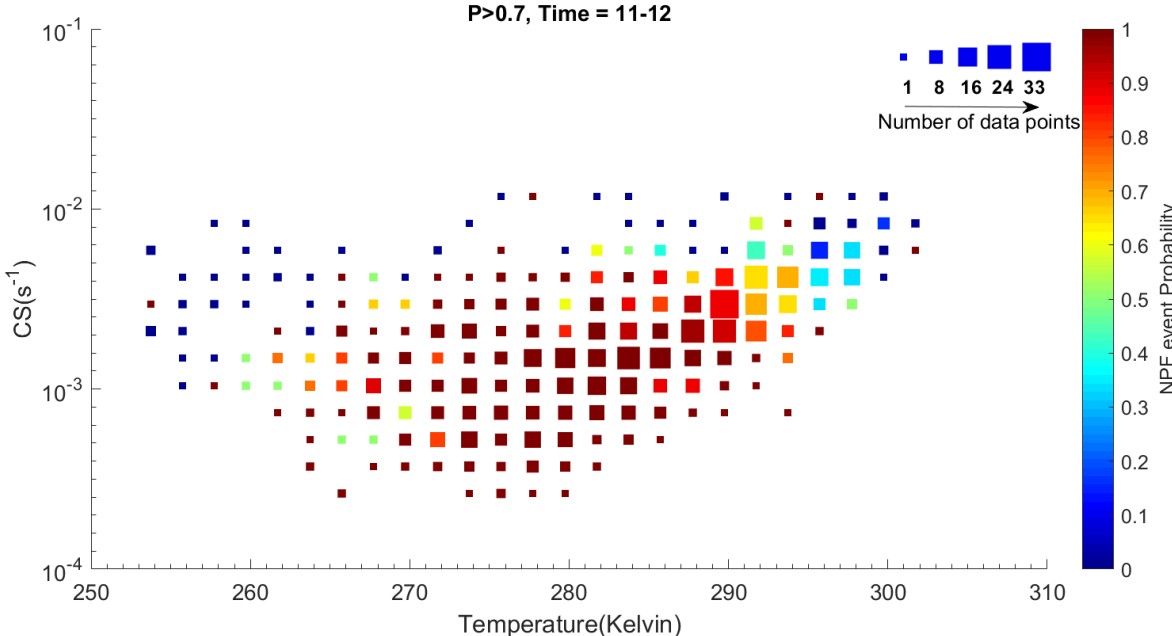


**Figure 13: NPF probability distribution based on the CS and temperature conditions during clear-sky days (11:00 -12:00).**
**Marker size indicates number of days included in the probability calculation within every cell.**