# Peer review of "Long-term analysis of clear-sky new particle formation events and non-events in Hyytiälä"

_Atmospheric Chemistry and Physics, 2016_

## Referee Comment (RC1) · Anonymous Referee #1 · 4 Nov 2016

Summary review

This manuscript uses a large dataset of nucleation events from the Hyytiala site to determine a criterion for new particle formation on days with low cloud cover, when most nucleation events at Hyytiala occur. The conclusion of the data analysis might be best paraphrased by "the probability of a new particle formation event on a clear-sky day in Hyytiala springtime is determined almost exclusively by the condensation sink on that day". It should be emphasised that very similar conclusions were reached before by Hyvonen et al in 2005, although with a less sophisticated analysis of a smaller dataset, and the text of the present manuscript could be made more coherent and complete. However, the analysis of the authors does nicely complement previous studies, the data are valuable and appropriately summarised by the plots that are presented, and, overall, the manuscript does represent useful new work.

The paper is within the scope of the journal, the title and abstract are reasonable, and due credit is generally given to previous studies. The quality of the written English is also generally fine. After the authors have improved the interpretation of the data in accordance with my comments, better justified the criterion for new particle formation, and added the full dataset as supplementary materials, I believe the paper would be appropriate for publication in ACP.

General comments

The research follows other studies considering the effect of cloud cover on NPF events (Baranizadeh et al in Boreal Env. Research) and several criteria for predicting new particle formation (papers by Kuang (ACP 10 8469), and Hyvonen, Nieminen as cited in the manuscript), some of which have also considered solar radiation levels. This research shows that cloudiness, condensation sink and temperature, when used together, can effectively predict the probability of a nucleation event at this important field station in spring, but rather less effectively in summer, autumn and winter.

The authors should be more explicit in motivating their work. Yes, aerosols are important for climate and nucleation is an important source of aerosols, but it should be explained more clearly why predicting the probability of a nucleation event in springtime at Hyytiala will help people improve their understanding of the atmosphere. For example, the authors could point out that Hyytiala is reasonably representative of semi-clean forested environments in the Northern Hemisphere, and that it is a suitable site without too many highly localised sources of aerosol that are difficult to model. And, since the authors present a criterion that is only effective in spring, explain that this is the most important season for NPF. And that 20 years of detailed observation data are not readily available at other sites.

The authors investigate several other variables that should be correlated to NPF event probability, but they do not explain why CS and temperature, and not the other variables, feature in their final criterion. For example, the box-and-whisker plots (where the

cloudiness parameter is than 0.7, indicating clear skies) do not clearly suggest that on clear sky days, T offers a better separation between NPF events and non-events than RH. The correlation matrix shows RH and CS are less correlated than T and CS, so RH might have more discriminating power. Moreover, RH on non-event days is almost always higher than on event days, while temperature on non-event days is higher in winter and spring but lower in autumn and winter. Therefore, in principle one might expect RH to be a better second variable than T when all seasons are considered, in line with Hyvonen et al, even after one has separated clear sky days and cloudy days as suggested by Baranizadeh et al already. The authors should quantitatively demonstrate why their criterion offers better discriminating power than a few other obvious possibilities, such as RH/CS.

In addition to the criterion for new particle formation events, the paper also aims to quantify the effect of cloudiness on NPF event frequency. However, this was done already by Baranizadeh et al and it should be made clearer what this manuscript adds to this relatively comprehensive previous work. The authors should either remove all but a very brief summary of this from the paper, or state clearly how their analysis relates to that of Baranizadeh et al with a sentence like "Our work confirms the conclusions of Baranizadeh et al with a complementary dataset".

The paper also aims to "find out the connection between nucleating precursor vapours and new particle formation rates". The authors should re-think this part of the manuscript. The analysis has the potential to provide interesting conclusions, but currently it is not well connected to the rest of the paper and the approach of the authors does not match the stated aim. The sentence I quote here is misleading because the connection is assumed by the manuscript, not "found": the new particle formation rates presented in the paper are not calculated from the rate of change of particle concentration, but from a parameterisation of the nucleating precursor vapour proxy concentrations. This is not a bad approach but just needs to be described more carefully: the comparison of the probability of a new particle formation event to the parameterised

nucleation rate is still a useful exercise.

To connect this to the rest of the paper, the authors could consider presenting this study as a comparison of the effectiveness of their condensation sink based criterion and their nucleation rate parameterisation at determining whether or not a nucleation event will occur on a given day. Then the conclusion might indicate explicitly that the parameterisation is a poor criterion for NPF compared to the condensation sink-temperature criterion (except perhaps in winter, from Figure 10?). Since parametrisations of this form might be considered reasonable starting points to determine whether or not NPF should occur (naively, a high parameterised rate ought to imply a nucleation event is likely), this would seem to be an interesting message. While Figure 10 is helpful in providing this message, further evidence could be obtained by re-plotting some of the data so that the criteria can be compared more directly: Figures 5a and 9a can be compared, but it would be better if the combined criterion including temperature were plotted on the y axis of a new version of Fig. 5a, since this should further improve the separation.

The careful statistical summaries presented in this manuscript do convince the reader that the underlying dataset is valuable. The large size allows statistically significant results to be extracted. A csv table (or similar) containing the full dataset, or at a minimum a list of dates studied over the 20 years with, on each day, the condensation sink, temperature, RH, cloudiness parameter, and whether or not the day contained a nucleation event, should be included in supplementary materials. While much of this information is already available, via the smartSMEAR website for example, a carefully compiled dataset specific to this paper would still be very useful. It would allow, for example, modellers comparing event frequency in models and observations to split up the dataset into individual years and compare model to measurements selected from the overall dataset to match their model simulations. A brief explanation of where subsets of the data have been published before should accompany the data file.

Specific comments
Summary: the sentence "This study serves as basis for scientists aiming at improving their understanding towards new particle formation" should be rephrased to improve the written English, for example "This study serves as a basis for scientists aiming to improve their understanding of new particle formation."

Abstract: "utilizing"->"building on" "In this comparison we considered, for example, the effects of calculated particle formation rates, condensation sink, trace gas concentrations and various meteorological quantities." -> considered the effect on what? "The formation rate of 1.5 nm particles was calculated by using proxies for gaseous sulfuric acid and oxidized products of low volatile organic compounds"-> add "and a nucleation rate parameterization" after "compounds" "As expected, our results indicate an increase in the frequency of NPF events under clear-sky conditions."-> "increase under clear-sky conditions compared to cloudy conditions"

"The calculated formation rate of 3 nm particles showed a notable difference between the NPF event and non-event days during clear-sky conditions, especially in winter and spring"-> so in cloudy conditions do you get high NPF rates but no events? Please be more explicit here.

Line 59: "That study" -> which of the three cited?

Line 88: The title of section 2.1.1 should be amended to make it clear that it is this section which explains how events are categorised, and this section should be extended with a very brief summary of how Dal Maso et al decide whether a day is an event, non-event or undefined day.

Line 127: At least four possible MT proxies are presented in Kontkanen et al. Which one did you use? Is it the recommended proxy MTproxy1,doy? Please specify.

Line 140-147: this section needs more detail on the data analysed and the characteristics of NPF in Hyytiala. Table 1 caption implies all of the data analysed are from the months of March to May, but this seems not to be true. However, it is clear that the

instruments would not be running every single day between 1 January 1996 and 31 December 2015. While Figure 2 is helpful here, it also needs some additional explanation and referencing early in the text. The brief statements about the seasonal cycle at lines 227-232 are confusing without this additional context.

Line 160: Somewhere here it would be good to state why you calculate J3, why not just use J1.5?

Line 174: Please summarise very briefly the improvement made by Kontkanen (2016).

Line 197: It would be helpful to state the number of undefined days here, so the reader does not wonder how it can be that 877 days are events, 229 are non-events, and 55% of days are event days.

Line 205 "days having less" -> "days with fewer"

Lines 204-209: Please rewrite or combine with the previous paragraph to ensure the message of this paragraph does not repeat the message of the previous paragraph

Line 213: "In order to find out clear results and conclusions, we will focus on comparison between NPF events and non-events in following sections."-> this is long-winded, could shorten to "Undefined events are not considered further in the analysis"

Line 230: If the annual trend is important to note, state explicitly what is the annual trend.

Line 235: What is the median and percentile of a trajectory? The median compass direction at the point on the trajectory where it arrives at Hyytiala, or the median compass direction of some kind of average over the length of the trajectory? Does "at every half hour" mean for the arrival of the air masses at Hyytiala every half hour or for one trajectory, moving back along it by half an hour at a time?

Line 252: "However, the monthly cycle of CS on non-event days had two maxima, one in spring and another one in autumn"- what is the reader meant to conclude from this

sentence?

Line 257: "The temperature at which clear-sky NPF events occurred was different for each month" - > The following sentences are not really 'examples' of this sentence. I would delete this.

Line 266: "even though it might also be attributed to the presumable increase" -> but it might also be attributed to the increase"

Line 279-281: this sentence needs a verb outside the "while" clause

Line 281: Increased RH leads to increased production of H2SO4. Additionally, even with constant H2SO4 concentrations, nucleation rates increase with RH in flow tube or chamber studies (e.g. Duplissy et al, JGR 2015) and are expected to from theory (Merikanto et al, JGR 2015, Vehkamaki et al 2002). However, it is indeed clear from Fig 5c that RH is negatively correlated with nucleation. This could be due to any number of reasons, but it seems odd to point out the Boy & Kulmala study on RH limiting VOC ozonolysis without discussing the far more robust and well-established evidence from atmospheric chemistry that RH should promote nucleation of sulphuric acid.

Line 303: Specify that the OxOrg proxy concentration depends on temperature via the MT proxy in Kontkanen et al. Also see previous comment concerning this proxy.

Line 323: The comparison between J1.5 and J3 is interesting but should be made more explicit – how much later is the peaking time of J3 than J1.5? Are Figures 8b and 9b the same, or are there differences? Would you expect differences, based on how long it takes particles to grow from 1.5 to 3nm in general?

Line 331 It is stated that figure 10 represents "median diurnal cycles". The description of what this actually means is currently a bit hard to follow, and it needs to be repeated in the figure caption. If I understand correctly, the median CS is calculated for each half hour, and plotted against the corresponding median J3 value. Perhaps it would be clearer to describe the plot by saying "the J3 and CS data were divided into 30 groups

according to the time of day at which the data were recorded, and the median J3 and CS values for each group were calculated. The first group of data were recorded between 5am and 5.30am, the next from 5.30am to 6am, and so on until 8pm local time" (with adjustments for the precise times/numbers of groups). The figure would also be clearer if the scales of the axes were better optimised so that the data extend closer to the extremes of the axis ranges.

Line 342. From Figure 10, it is interesting that in summer, autumn and winter the highest J3 on non-event days is almost as high as on event days, and one would therefore expect the J1.5 to be very similar to the J1.5 on event days. This is in sharp contrast to the large differences between event and non-event days shown in Figure 9b for spring. For autumn, winter and summer, figure 10 would imply that the J rate by itself is a poor predictor for whether or not an event will occur. This is surely a useful message for your paper: it could be used to emphasise the importance of your new discriminating variable, based on condensation sink, which from Figure 10 clearly should perform better than the nucleation rate, which naively sounds like a more obvious variable to determine whether or not a nucleation event is occurring.

Figure 2: In addition to the helpful rows of numbers presented below the box plot, it would be helpful to state the number of event and non-event days with P > 0.7 in two additional rows.

Figure 12: why does the criterion for NPF you have developed perform badly in summer, autumn and winter?

It would be possible to determine quantitatively the benefit of the clear-sky classification by applying the NPF criterion in the clear-sky case and also without first separating clear sky events from non-events. Please state the effectiveness of the criterion in the case where you do not distinguish clear-sky and cloudy events, in order to prove the usefulness of the clear-sky distinction by showing that the NPF criterion is less effective without it.

References: should cite Kuang et al, ACP 2010, "An improved criterion for new particle formation in diverse atmospheric environments" somewhere

Table 1: +/-0.45 does not really indicate "high correlation": for this description to be justified I think you need +/-0.7 at least! Also, since the tables are symmetric about the diagonal, please remove the lower triangle (or replace with "-") so the reader does not have to check the upper and lower triangles are the same.

Figures 1/2: what is the "relevant statistical limit"?

Figure 3 caption: "5.4%, (add comma) making the classification biased." Please state more explicitly what you mean here : do you have only global radiation data for 5.4% of the days in 1998?

[Figure]

---

## Referee Comment (RC2) · Anonymous Referee #2 · 6 Dec 2016

In their manuscript, the authors present an in-depth analysis of a long dataset of aerosol, meteorology, trace gas and irradiation measurements at the SMEAR station in Finland. The analysis is performed to find the key parameters that would explain new particle formation.

Similar analyses with the same datasets have already been performed several times, as explained by the authors. However, in this analysis the authors focus on eliminating the effect of cloudiness in the analysis, which is an interesting approach and merits publication in ACP. The data aquisition methods are described in good details, and the data analysis mostly follows the procedures described in the cited literature. Some of the specific methods for this paper could be described in more detail and the choices and justification for them should be explained in the text (see detailed comments).

[Figure]

A similar analysis without the cloudiness parameter has been performed earlier, it would be nice to see a direct comparison of the analysis of regarding the separation of events and non-events (Hyvönen et al., 2005). It should be quite straightforward to perform the same linear discriminant analysis as the Hyvönen paper for the CS and RH data (Fig 4 in the Hyvönen et al paper), and compare whether the result has changed.

Also, I think it should be made clear that the event probability described in Figure 13 and in section 3.3.4 is different from the equation 6, and also different from the event probability introduced in the Hyvänen et al paper. In the latter, the event probablility is computed from the LDA analysis, while in the current paper the probability seems to be directly calculated from data, and thus it is not a predictive equation. I suggest that the authors revise this part of the paper. Also, if no real propability-giving predictive equation is given, I think that aim IV in the Introduction (line 66) should be revised. However, overall I think that the paper is a potentially good addition to the literature of understanding NPF, and its topic is certainly appropriate for ACP. Therefore, if the above corrections and the detailed notes given below can be considered by the authors, I would suggest publication. The corrections and revisions are, in my opinion, minor.

Detailed notes:

line 150-158 and 223-225: I do not fully understand the definition of the clear-sky day presented by the authors. Generally, it is known that particle formation occurs around noon, and that especially the mixing of the residual layer in the morning seems to have an influence. From that, I can follow that using the morning value is useful in the analysis. However, only the median P value for three hours is used. This raises the following questions:

* Were only events that started during this three-hour window included in the analysis?
* Why was the median used? In this case, a time period that is for example 1 hour 29 minutes cloudy and 1 hour 31 minutes sunny gets classified as a sunny (clear-sky) day. Does the result change when the mean is used? * what is the basis of using the value

0.7?

The reasoning between this central point in the methodology should be explained in much more detail, as I expect that similar analyses will be performed in the future for other sites, and therefore the method should be as robustly implemented as possible. Also, can the authors give insight on how sensitive the method is on the limit value of P chosen?

Line 198: ". . . radiation is essential for NPF as these events occur mainly during daylight hours." If radiation was essential, no NPF could be observed during nighttime. In the literature, several examples of NPF during nighttime can be found. Please rephrase.

line 200: is SA really the main component of freshly formed particles? If heteromolecular nucleation is the prevailing mechanism, the the organic compound is as important. Both are still likely to be formed photochemically, so I think that this sentence can be fixed by just by rewording (e.g. '..because the main components of freshly formed particles are likely formed photochemically. . .')

line 235-245: Please clarify also in the text and in the caption of Figure 4 that these results refer to clear-sky events only.

Line 251-254: As the CS is highest for event days, but not so for non-event days, does the presented conclusion that the CS is the reason for the minimum in events in summer really follow? It seems to me that in summer, events may occur despite high CS, and the actual reason for non-events is not the inhibiting effect of CS. If the authors disagree, this could be clarified.

Line 270: with monthly I think that the authors mean yearly

Line 280-281, '. . .low or almost no correlation. . .' something seems to be missing in this sentence.

lines 331-350: I don't really understand what is shown in figure 10, and therefore also

don't follow the explanation in this paragraph. What is meant by diurnal cycle here? By definition it means a repeating pattern that occurs every 24 hours, and I don't see how this could result in Figure 10. Please clarify and rewrite, or replace with the correct figure.

Line 357: The procedure of finding the separating line in Fig 11-12 is described very poorly. Is this done by linear discriminant analysis (such as e.g. in the cited Hyvönen et al., (2005) paper or some other method? The authors should describe this in more detail. I'm especially concerned about the sentence "the data points have been estimated by taking the non-events with the Âălowest possible CS which still fit the linear separation"; was some kind of data selection applied to produce the figure?

Figures: Several figures have the sentence "The lines extending 1.5 times from the central box represent the remaining of the data yet still within the relevant statistical limit. " Please clarify what this means: firstly, what is 1.5 times from the central box (the lines seem to have different lengths, eg. in fig. 5. Also, clarify what is meant by relevant statistical limit.
* * *

---

## Author Comment (AC1) · 3 Mar 2017

**Reply to Referee #1**

We thank Referee #1 for their helpful suggestions. We replied to the comments below. The bold text refers to the referee's comments, and the text in italics are additions to the manuscript. The line numbers mentioned in the text below refer to the ACPD version of the manuscript.

**I.     Underline{General comments}**

I.     General comments

**1.     The research follows other studies considering the effect of cloud cover on NPF events (Baranizadeh et al in Boreal Env. Research) and several criteria for predicting new particle formation (papers by Kuang (ACP 10 8469), and Hyvonen, Nieminen as cited in the manuscript), some of which have also considered solar radiation levels. This research shows that cloudiness, condensation sink and temperature, when used together, can effectively predict the probability of a nucleation event at this important field station in spring, but rather less effectively in summer, autumn and winter. The authors should be more explicit in motivating their work. Yes, aerosols are important for climate and nucleation is an important source of aerosols, but it should be explained more clearly why predicting the probability of a nucleation event in springtime at Hyytiala will help people improve their understanding of the atmosphere. For example, the authors could point out that Hyytiala is reasonably representative of semiclean forested environments in the Northern Hemisphere, and that it is a suitable site without too many highly localised sources of aerosol that are difficult to model. And, since the authors present a criterion that is only effective in spring, explain that this is the most important season for NPF. And that 20 years of detailed observation data are not readily available at other sites.**

We agree with the reviewer that it is important to mention the characteristics of Hyytiälä that make the location interesting and important for studying NPF. Accordingly, the following has been added to the manuscript (Line 55):

*The Station for Measuring Forest Ecosystem-Atmosphere Relations (SMEAR II) located in Hyytiälä, southern Finland, compiles up to 21 years of particle number size distribution and extensive complementary data, providing the longest size distribution time series in the world, and hence allows for robust NPF analysis which is not readily possible at other sites. The station is located in a homogenous Scots pine forest far from major pollution sources. Hyytiälä, thus, is classified as a background site representative of the semi-clean northern hemisphere boreal forests.*

Our focus on springtime is explained more thoroughly following the reviewer's suggestion (line 235):

*Since NPF is most frequent in spring, we dedicate our focus on this season (Figure 3a).*

**2.     The authors investigate several other variables that should be correlated to NPF event probability, but they do not explain why CS and temperature, and not the other variables, feature in their final criterion. For example, the box-and-whisker plots (where the cloudiness parameter is than 0.7, indicating clear skies) do not clearly suggest that on clear sky days, T offers a better separation between NPF events and non-events than RH. The correlation matrix shows RH and CS are less correlated than T and CS, so RH might have more discriminating power. Moreover, RH on non-event days is almost always higher than on event days, while temperature on non-event days is higher in winter and spring but lower in autumn and winter. Therefore, in principle one might expect RH to be a better second variable than T when all seasons are considered, in line**

**with Hyvonen et al, even after one has separated clear sky days and cloudy days as suggested by Baranizadeh et al already. The authors should quantitatively demonstrate why their criterion offers better discriminating power than a few other obvious possibilities, such as RH/CS.**

Looking at the median values presented in RH monthly box plots in Figure 5c, might give the idea that RH values are clearly separated between events and non-events. Considering the wider spread in the RH data (25%-75% percentiles) for events and nonevents as seen in the boxplots, we feel this parameter is less conclusive in separating the two classes. However, based on the reviewer's suggestion, we plot RH vs CS (spring time window 11:00-12:00) below and compare it to T vs CS (spring time window 11:00-12:00) plot. The line in the figure is the Linear Discriminant Analysis (LDA) at 95% confidence that all the nonevent points are outside the line (to the right). The plots show that RH does not result in better separation than temperature (events from nonevents) as CS sink seems to be the main controlling factor. We then conclude that during clear-sky conditions the results are somewhat different from what Hyvönen et al. (2005) who did not consider clear-sky conditions only. Based on the aforementioned results, and following the reviewer's suggestion, we add the following to the text to line 369:

*Furthermore, we analyzed the effect of RH in separating the events from nonevents, similar to the study done on RH by Hyvönen et al. 2005. We found that compared with CS vs temperature data, depicting CS vs RH (data not presented) did not work better in separating NPF events from non-events during clear-sky conditions.*

[Figure]

**3.      In addition to the criterion for new particle formation events, the paper also aims to quantify the effect of cloudiness on NPF event frequency. However, this was done already by Baranizadeh et al and it should be made clearer what this manuscript adds to this relatively comprehensive previous work. The authors should either remove all but a very brief summary of this from the paper, or state clearly how their analysis relates to that of Baranizadeh et al with a sentence like "Our work confirms the conclusions of Baranizadeh et al with a complementary dataset".**

We added the suggested sentence to the manuscript to confirm the parallel between our study and Baranizadeh et al. 2014 to line (393): *Our work confirms, with a complementary dataset, the conclusions of Baranizadeh et al. (2014) that NPF events and non-events are typically associated with clear-sky and cloudy conditions, respectively.*

**4.       The paper also aims to "find out the connection between nucleating precursor vapours and new particle formation rates". The authors should re-think this part of the manuscript. The analysis has the potential to provide interesting conclusions, but currently it is not well connected to the rest of the paper and the approach of the authors does not match the stated aim. The sentence I quote here is misleading because the connection is assumed by the manuscript, not "found": the new particle formation rates presented in the paper are not calculated from the rate of change of particle concentration, but from a parameterisation of the nucleating precursor vapour proxy concentrations. This is not a bad approach but just needs to be described more carefully: the comparison of the probability of a new particle formation event to the parameterized nucleation rate is still a useful exercise.**

**5.**

We modified the aim mentioned by the referee into the following form:

*iii) Explore the connections between new particle formation rates calculated from precursor vapor proxies and the occurrence of NPF events.*

**6.       To connect this to the rest of the paper, the authors could consider presenting this study as a comparison of the effectiveness of their condensation sink based criterion and their nucleation rate parameterisation at determining whether or not a nucleation event will occur on a given day. Then the conclusion might indicate explicitly that the parameterisation is a poor criterion for NPF compared to the condensation sink-temperature criterion (except perhaps in winter, from Figure 10?). Since parametrisations of this form might be considered reasonable starting points to determine whether or not NPF should occur (naively, a high parameterised rate ought to imply a nucleation event is likely), this would seem to be an interesting message. While Figure 10 is helpful in providing this message, further evidence could be obtained by re-plotting some of the data so that the criteria can be compared more directly: Figures 5a and 9a can be compared, but it would be better if the combined criterion including temperature were plotted on the y axis of a new version of Fig. 5a, since this should further improve the separation.**

According to the suggestion of both reviewers and to clarify the message of Figure 10, we re-plot figure 10 into a diurnal cycle showing the median CS and the parameterized formation rate. The replacement aids the reader in visualizing the influence of the CS on $J_{3,C}$ as well as the negative correlation between the two. The plot will also improve and connect the parts of the paper together as the reviewer implied. Accordingly, we modify the related text in the manuscript to the following (Line 342):

*On NPF event days, the median approximated formation rate of 3 nm particles had its maximum value at about midday and was significantly higher than on non-events days (Figures 9b and 10). A clear negative relation could be seen between the median seasonal diurnal cycles of CS and $J_{3,C}$ on NPF event days (specifically during spring daytime) (Figure 10). This kind of relation was not observed during non-event days when these two quantities seemed to be independent of each other (Figure 10). In summer, the median value of $J_{3,C}$ was roughly similar between NPF events and non-events, whereas the median value of CS was almost ten times higher during the non-event days compared with event days. The high values of $J_{3,C}$ for the non-event days in summer, despite the high CS values, seem to suggest that some other factor limits the actual NPF rate. One option is that the freshly formed clusters are rapidly evaporated due to higher ambient temperatures (see Fig. 5b). This will be discussed in more detail in the following section.*

**7.     The careful statistical summaries presented in this manuscript do convince the reader that the underlying dataset is valuable. The large size allows statistically significant results to be extracted. A csv table (or similar) containing the full dataset, or at a minimum a list of dates studied over the 20 years with, on each day, the condensation sink, temperature, RH, cloudiness parameter, and whether or not the day contained a nucleation event, should be included in supplementary materials. While much of this information is already available, via the smartSMEAR website for example, a carefully compiled dataset specific to this paper would still be very useful. It would allow, for example, modellers comparing event frequency in models and observations to split up the dataset into individual years and compare model to measurements selected from the overall dataset to match their model simulations. A brief explanation of where subsets of the data have been published before should accompany the data file.**

As the reviewer mentioned, the data used for our analysis can be downloaded from smartSMEAR. Because compiling 20 years of data results in a massive file, we do not think that it is necessary. However, the authors are happy to collaborate and send the needed data to modelers to improve or add valuable data to the literature.

**II.     Specific comments**

**Summary: the sentence "This study serves as basis for scientists aiming at improving their understanding towards new particle formation" should be rephrased to improve the written English, for example "This study serves as a basis for scientists aiming to improve their understanding of new particle formation."**

**1.     Abstract: "utilizing"->"building on".**

We replaced the word utilizing by building on.

**2.     "In this comparison we considered, for example, the effects of calculated particle formation rates, condensation sink, trace gas concentrations and various meteorological quantities." -> considered the effect on what?**

We reformulated the sentence according to the reviewer's suggestion in line 20 by adding the following continuation of the sentence: *in discriminating NPF events from non-events*.

**3.     "The formation rate of 1.5 nm particles was calculated by using proxies for gaseous sulfuric acid and oxidized products of low volatile organic compounds"-> add "and a nucleation rate parameterization" after "compounds".**

We added: "*and a nucleation rate parameterization factor*" to line 22.

**4.     "As expected, our results indicate an increase in the frequency of NPF events under clear-sky conditions."-> "increase under clear-sky conditions compared to cloudy conditions".**

We added the suggested to the sentence "*in comparison to cloudy ones*" to line 23.

**5.     "The calculated formation rate of 3 nm particles showed a notable difference between the NPF event and non-event days during clear-sky conditions, especially in winter and spring"-> so in cloudy conditions do you get high NPF rates but no events? Please be more explicit here.**

The objective of the paper is to select the clear-sky days and compare the events and nonevents within this dataset and we only discuss the clear-sky days in this article. As this is implicit based on the previous sentences, we removed the "during clear-sky conditions" from the sentence in order not to cause confusion such as for the

referee. The sentence quoted by the reviewer above means that within the selected clear-sky dataset, the formation rate between events and non-events is different. However, for the interest of the reviewer, we present below a table of statistical analysis of formation rates at 1.5nm between clear-sky (P>0.7) and cloudy (P<0.3) events and non-events.

| $J_{1.5}$ (cm$^{-3}$ s$^{-1}$) | 5th percentile | 25th percentile | Median | 75th percentile | 95th percentile |
|---|---|---|---|---|---|
| Clear-sky events | 0.10 | 0.56 | 1.37 | 2.89 | 8.76 |
| Clear-sky non-events | 0.03 | 0.28 | 0.77 | 1.82 | 5.72 |
| Cloudy events | 0.01 | 0.21 | 0.72 | 1.93 | 7.50 |
| Cloudy nonevents | 0.01 | 0.06 | 0.17 | 0.46 | 2.00 |

Results appear as expected as the $J_{1.5}$ is calculated directly from the concentrations of sulfuric acid and OxOrg (equation 5) which are both produced photochemically. Thus, it is, as shown, expected that the parametrized formation rate is higher in clear-sky conditions than cloudy ones, in general. The pattern is the same for the $J_3$. However, in both cases, particle formation rate is higher on event days than non-events if we take either clear-sky or cloudy conditions, separately.

**6.        Line 59: "That study" -> which of the three cited?**

The sentence "That study" was replaced with: "They" for clarity. The Baranizadeh et al. are the only authors being actively addressed in the previous sentence.

**7.        Line 88: The title of section 2.1.1 should be amended to make it clear that it is this section which explains how events are categorised, and this section should be extended with a very brief summary of how Dal Maso et al decide whether a day is an event, non-event or undefined day.**

The title of section 2.1.1 is modified to: *"New Particle Formation Events Classification"*.

The following is added to this section: *"The latter uses a decision criterion based on the presence of particles < 25 nm in diameter and their consequent growth to Aitken mode. Event days are days on which sub 25 nm particle formation and growth are observed. Non-event days are days on which neither modes are present. Undefined days are the days which do not fit either criterion."*

**8.        Line 127: At least four possible MT proxies are presented in Kontkanen et al. Which one did you use? Is it the recommended proxy MTproxy1,doy? Please specify.**

We added the missing information to the manuscript.

**9.        Line 140-147: this section needs more detail on the data analysed and the characteristics of NPF in Hyytiala. Table 1 caption implies all of the data analysed are from the months of March to May, but this seems not to be true. However, it is clear that the instruments would not be running every single day between 1 January 1996 and 31 December 2015. While Figure 2 is helpful here, it also needs some additional explanation and referencing early in the text. The brief statements about the seasonal cycle at lines 227-232 are confusing without this additional context.**

While the analysis included in the manuscript is comprehensive and includes data between 1996 and 2015, some of the figures/tables include only part of the data. For instance, as the reviewer pointed out that Table 1 relates only to the spring time correlation. The table includes only spring time as there are the months with the highest

frequency of events. Including all months in one correlation calculation would not give reasonable results. The number of classified clear-sky data points are included in Figure 2.

**10.      Line 160: Somewhere here it would be good to state why you calculate J3, why not just use J1.5?**

In previous studies which did not consider clear-sky conditions, it has been observed that there is a clear difference in $J_3$ between event and non-event days while $J_{1.5}$ is more similar (Kulmala et al. 2013). Furthermore, we have more direct $J_3$ measurements to which we can compare the calculated values. Classification of events around the world are based on DMPS data at 3 nm and above, rather than on data below 3 nm diameter.

Line 320 now reads*: "Since previous studies have shown that there is a clear difference in $J_3$ between the event and non-event days, and much less difference in $J_{1.5}$ (Kulmala et al. 2013), we decided to focus on $J_3$ in our event to non-event discrimination."*

**11.      Line 174: Please summarise very briefly the improvement made by Kontkanen (2016).**
The main improvements in OxOrg proxy by Kontkanen et al. (2016) compared to the previous version (Lappalainen et al., 2009) of the proxy are: 1) Monoterpene concentrations measured during the whole day were used for the proxy. 2) The mixing within the boundary layer, diluting monoterpene concentration, was considered. (3) The oxidation of monoterpenes by nitrate radical ($NO_3$) was included.

The explanation of the improvements by Kontkanen et al. 2016 are discussed in the method's section 2.1.4 lines 122-127.

**12.      Line 197: It would be helpful to state the number of undefined days here, so the reader does not wonder how it can be that 877 days are events, 229 are non-events, and 55% of days are event days.**

Based on the reviewer's suggestion we add the missing number of undefined days to line 198. "*with 877 days classified as NPF events, 560 undefined days and only 229 as non-events."*

**13.      Line 205 "days having less" -> "days with fewer"**

See following comment.

**14.      Lines 204-209: Please rewrite or combine with the previous paragraph to ensure the message of this paragraph does not repeat the message of the previous paragraph**

We modified the text in lines 204-209 based on the reviewer's suggestion.

**15.      Line 213: "In order to find out clear results and conclusions, we will focus on comparison between NPF events and non-events in following sections."-> this is long-winded, could shorten to "Undefined events are not considered further in the analysis"**

We considered the reviewer's suggestion.

**16.      Line 230: If the annual trend is important to note, state explicitly what is the annual trend.**

As stated in the next sentences, there was no clear trend, but the number of events varied from year to year. We modified the sentence to: *The total number of NPF events varied from year to year between 1996 and 2015.*

**17.      Line 235: What is the median and percentile of a trajectory? The median compass direction at the point on the trajectory where it arrives at Hyytiala, or the median compass direction of some kind of average over the length of the trajectory? Does "at every half hour" mean for the arrival of the air masses at Hyytiala every half hour or for one trajectory, moving back along it by half an hour at a time?**

Based on the reviewer's comment and for clarity we modified the section explaining the median and percentile of the trajectories. *The medians and similarly the percentiles were calculated by taking the median compass direction at every point on the trajectory (1 hour between every two points), arriving every half an hour at Hyytiälä.*

**18.      Line 252: "However, the monthly cycle of CS on non-event days had two maxima, one in spring and another one in autumn"- what is the reader meant to conclude from this sentence?**

We added a continuation to the sentence in line 252 to ensure clarity: *However, the monthly cycle of CS during non-event days had two maxima, one in spring and another one in autumn, which might suggest that during these seasons, high values of CS prevented NPF to occur during those particular days.*

**19.      Line 257: "The temperature at which clear-sky NPF events occurred was different for each month" - > The following sentences are not really 'examples' of this sentence. I would delete this.**

We endorse the reviewer's suggestion.

**20.      Line 266: "even though it might also be attributed to the presumable increase" -> but it might also be attributed to the increase"**

We did the change.

**21.      Line 279-281: this sentence needs a verb outside the "while" clause**

See the following comment.

**22.      Line 281: Increased RH leads to increased production of H2SO4. Additionally, even with constant H2SO4 concentrations, nucleation rates increase with RH in flow tube or chamber studies (e.g. Duplissy et al, JGR 2015) and are expected to from theory (Merikanto et al, JGR 2015, Vehkamaki et al 2002). However, it is indeed clear from Fig 5c that RH is negatively correlated with nucleation. This could be due to any number of reasons, but it seems odd to point out the Boy & Kulmala study on RH limiting VOC ozonolysis without discussing the far more robust and well-established evidence from atmospheric chemistry that RH should promote nucleation of sulphuric acid.**

Although increased RH leads to enhanced production of $H_2SO_4$ even with constant $H_2SO_4$ concentrations, nucleation rates increase with RH in flow tube or chamber studies (e.g.Duplissy and Flagan, 2016) and are expected to from theory (Merikanto et al., 2016;Vehkamäki et al., 2002). Our results show that RH is negatively correlated with nucleation, mainly because pure $H_2SO_4$ nucleation is not expected in Hyytiälä. However, in correspondence to the reviewer's suggestions we add more details to the paragraph referring to the effect of RH on NPF:

*Previous studies in Hyytiälä have found that events are accompanied with lower RH values in comparison to non-events (Hamed et al., 2011). Other studies have proposed that increased RH limits some VOC (Volatile Organic Compounds) ozonolysis reactions, preventing the formation of come condensable vapors necessary for nucleation (Boy and Kulmala, 2002). This might partially explain the observed anti-correlation between RH and particle formation rates. Therefore, it seems plausible that RH affects NPF via atmospheric chemistry rather than via changing the sink term for condensing vapors and small clusters. Additionally, we found clear differences in how trace gas concentrations were associated with RH between the NPF event and non-event days (Table 1). For instance, $O_3$ showed a strong negative correlation with RH during events and non-events. However, during non-event days, a positive correlation appears between RH and each of CO, $SO_2$ and NOx while the correlation between those seems to be absent during event days. Our results show that air masses coming from central Europe and passing over the Baltic Sea tend to have higher values of RH.*

**23.    Line 303: Specify that the OxOrg proxy concentration depends on temperature via the MT proxy in Kontkanen et al. Also see previous comment concerning this proxy.**

A brief summary of the derivation of the OxOrg proxy concentration is present on lines 122 – 127.

**24.    Line 323: The comparison between J1.5 and J3 is interesting but should be made more explicit – how much later is the peaking time of J3 than J1.5? Are Figures 8b and 9b the same, or are there differences? Would you expect differences, based on how long it takes particles to grow from 1.5 to 3nm in general?**

Differences could arise also if the growth from 1.5-3nm is the critical step for NPF, rather than the initial clustering forming 1.5nm particles, as is suggested by some studies (Kulmala et al., 2013). During clear-sky conditions, Table below summarizes the time delays until 3 nm particles are formed. Time delay is calculated as $d_{dp}$/GR. The peak times varies thus based on the GR and differs on each day. We modify the text in the manuscript (Line 329) to include the discussion related to the delay. Figure 9 is also replaced with the corrected one which includes the delay.

| Time Delay | Median |
|------------|--------|
| Events | 0.6 h |
| Non-events | 0.4 h |

'*On event days, in comparison to springtime $J_{1.5,C}$ which peaked at around 10:45 (Figure 8b), $J_{3,C}$ peaked typically about half-an hour later. This time delay indicates how long it takes for the particles grow from 1.5 nm to 3 nm. This growth is a critical step of NPF (Kulmala et al. 2013), and depends on concentrations of available vapour precursors.*'

**25.     Line 331** It is stated that figure 10 represents "median diurnal cycles". The description of what this actually means is currently a bit hard to follow, and it needs to be repeated in the figure caption. If I understand correctly, the median CS is calculated for each half hour, and plotted against the corresponding median J3 value. Perhaps it would be clearer to describe the plot by saying "the J3 and CS data were divided into 30 groups according to the time of day at which the data were recorded, and the median J3 and CS values for each group were calculated. The first group of data were recorded between 5am and 5.30am, the next from 5.30am to 6am, and so on until 8pm local time" (with adjustments for the precise times/numbers of groups). The figure would also be clearer if the scales of the axes were better optimised so that the data extend closer to the extremes of the axis ranges.

Figure 10 was modified into a clearer version. See comment 4 in the General comments.

**26.     Line 342.** From Figure 10, it is interesting that in summer, autumn and winter the highest J3 on non-event days is almost as high as on event days, and one would therefore expect the J1.5 to be very similar to the J1.5 on event days. This is in sharp contrast to the large differences between event and non-event days shown in Figure 9b for spring. For autumn, winter and summer, figure 10 would imply that the J rate by itself is a poor predictor for whether or not an event will occur. This is surely a useful message for your paper: it could be used to emphasise the importance of your new discriminating variable, based on condensation sink, which from Figure 10 clearly should perform better than the nucleation rate, which naively sounds like a more obvious variable to determine whether or not a nucleation event is occurring.

Figure 10 is modified into a clearer version as well as the text accompanying it. See comment 4 in the General comments.

**27.     Figure 2:** In addition to the helpful rows of numbers presented below the box plot, it would be helpful to state the number of event and non-event days with P > 0.7 in two additional rows.

Based on the suggestion, we added the corresponding data to Figure 2 and modified the figure caption.

**28.     Figure 12:** why does the criterion for NPF you have developed perform badly in summer, autumn and winter? It would be possible to determine quantitatively the benefit of the clear-sky classification by applying the NPF criterion in the clear-sky case and also without first separating clear sky events from non-events. Please state the effectiveness of the criterion in the case where you do not distinguish clear-sky and cloudy events, in order to prove the usefulness of the clear-sky distinction by showing that the NPF criterion is less effective without it.

The differences between meteorological parameters is clearly lower in all other seasons in comparison to spring, which explains the limitation of our criterion. Although precursor vapors have high concentrations in summer, however, the concentrations are similar between event and non-event days which makes it hard to separate days into events and non-events based on vapor concentrations. We plotted below figure 12 (left) accompanied by a similar figure where the clear-sky classification had not been taken into account. In total, the number of total events and non-events were 1458 and 2118, respectively, while in clear-sky conditions the numbers were 877 and 229, respectively. Accordingly, and as shown in the plot below, if no clear-sky selection is done, it is

basically almost impossible to separate the events from the non-events. While in clear-sky conditions, it was possible to very well separate events from the non-events in spring, which makes clear-sky distinction useful.

[Figure]

**29.** **References: should cite Kuang et al, ACP 2010, "An improved criterion for new particle formation in diverse atmospheric environments" somewhere**

Kuang et al. (2010) developed a criterion for NPF probability based on a dimensionless parameter (ratio of particle loss rate to growth rate), which determines when the newly formed clusters are likely to grow to detectable sizes. They conclude this criterion to work in diverse environments, however, they did not explore the dependency of their parameter on atmospheric conditions. Line 383 now reads "Although previous studies have developed criteria for NPF probability which could work in diverse environments (Kuang et al., 2010), they did not explore the dependency of their criterion on atmospheric conditions."

**30.** **Table 1: +/-0.45 does not really indicate "high correlation": for this description to be justified I think you need +/-0.7 at least! Also, since the tables are symmetric about the diagonal, please remove the lower triangle (or replace with "-") so the reader does not have to check the upper and lower triangles are the same.**

We changed the coloring criteria in Table 1 so that only values higher than 0.7 are indicated as high.

**31.** **Figures 1/2: what is the "relevant statistical limit"?**

The default box plots refer to 99.3% statistical limit. For clarity the figure caption accompanying all boxplots is modified to: *The lines extending from the central box represent 1.5 x interquartile range which includes 99.3% of the data inclusive. Data outside this statistical limit are considered outliers and are marked with red crosses.*

**32.** **Figure 3 caption: "5.4%, (add comma) making the classification biased." Please state more explicitly what you mean here: do you have only global radiation data for 5.4% of the days in 1998?**

We did the change.

**References**

Boy, M., and Kulmala, M.: Nucleation events in the continental boundary layer: Influence of physical and meteorological parameters, Atmospheric Chemistry and Physics, 2, 1-16, 2002.

Duplissy, J., and Flagan, R.: Effect of ions on sulfuric acid-water binary particle formation: 2. Experimental data and comparison, Journal of Geophysical Research. Atmospheres, 121, 1752-1775, 2016.

Hamed, A., Korhonen, H., Sihto, S. L., Joutsensaari, J., Järvinen, H., Petäjä, T., Arnold, F., Nieminen, T., Kulmala, M., and Smith, J. N.: The role of relative humidity in continental new particle formation, Journal of Geophysical Research: Atmospheres, 116, 2011.

Hyvönen, S., Junninen, H., Laakso, L., Maso, M. D., Grönholm, T., Bonn, B., Keronen, P., Aalto, P., Hiltunen, V., and Pohja, T.: A look at aerosol formation using data mining techniques, Atmospheric Chemistry and Physics, 5, 3345-3356, 2005.

Kuang, C., Riipinen, I., Sihto, S.-L., Kulmala, M., McCormick, A., and McMurry, P.: An improved criterion for new particle formation in diverse atmospheric environments, Atmospheric Chemistry and Physics, 10, 8469-8480, 2010.

Kulmala, M., Kontkanen, J., Junninen, H., Lehtipalo, K., Manninen, H. E., Nieminen, T., Petäjä, T., Sipilä, M., Schobesberger, S., and Rantala, P.: Direct observations of atmospheric aerosol nucleation, Science, 339, 943-946, 2013.

Lappalainen, H., Sevanto, S., Bäck, J., Ruuskanen, T., Kolari, P., Taipale, R., Rinne, J., Kulmala, M., and Hari, P.: Day-time concentrations of biogenic volatile organic compounds in a boreal forest canopy and their relation to environmental and biological factors, Atmospheric Chemistry and Physics, 9, 5447-5459, 2009.

Merikanto, J., Duplissy, J., Määttänen, A., Henschel, H., Donahue, N. M., Brus, D., Schobesberger, S., Kulmala, M., and Vehkamäki, H.: Effect of ions on sulfuric acid-water binary particle formation: 1. Theory for kinetic- and nucleation-type particle formation and atmospheric implications, J. Geophys. Res. Atmos, 121, 1736-1751, 2016.

Vehkamäki, H., Kulmala, M., Napari, I., Lehtinen, K. E., Timmreck, C., Noppel, M., and Laaksonen, A.: An improved parameterization for sulfuric acid–water nucleation rates for tropospheric and stratospheric conditions, Journal of Geophysical Research: Atmospheres, 107, 2002.

---

## Author Comment (AC2) · 3 Mar 2017

**Reply to Referee #2**

We thank Referee #2 for their helpful suggestions. We replied to the comments below. The bold text refers to the referee's comments, and the text in italics are additions to the manuscript. The line numbers mentioned in the text below refer to the ACPD version of the manuscript.

    I.        General comments:

**In their manuscript, the authors present an in-depth analysis of a long dataset of aerosol, meteorology, trace gas and irradiation measurements at the SMEAR station in Finland. The analysis is performed to find the key parameters that would explain new particle formation.**

**Similar analyses with the same datasets have already been performed several times, as explained by the authors. However, in this analysis the authors focus on eliminating the effect of cloudiness in the analysis, which is an interesting approach and merits publication in ACP. The data aquisition methods are described in good details, and the data analysis mostly follows the procedures described in the cited literature. Some of the specific methods for this paper could be described in more detail and the choices and justification for them should be explained in the text (see detailed comments).**

**1.      A similar analysis without the cloudiness parameter has been performed earlier, it would be nice to see a direct comparison of the analysis of regarding the separation of events and non-events (Hyvönen et al., 2005). It should be quite straightforward to perform the same linear discriminant analysis as the Hyvönen paper for the CS and RH data (Fig 4 in the Hyvönen et al paper), and compare whether the result has changed.**

Our results show that although the relative humidity seems to be a variable that influences NPF, when only clear-sky conditions are considered, the variation of RH between events and non-events does not seem to explain the occurrence of NPF. For instance, looking at figure 5c, we notice that although there might seem to be a difference in the median value of the RH when comparing event days and non-event days within each month, the percentiles seem to coincide minimizing the overall separation. However, we agree with the reviewer that it is important to compare with the suggested publication. Accordingly, we plotted clear-sky RH vs CS (below). The plots include spring clear-sky events and non-events within the time window 11:00-12:00 which has been proven shown to be the peak time of NPF Figures 8b and 9b. We also performed Linear Discriminant Analysis (LDA) according to Hyvönen *et al.* 2005 and added the 100% confidence level line to the plot. The 100% confidence limit corresponds to separating 95% of the non-events (to the right of the line in this case). We compare the corresponding spring RH vs CS plot with that of Temperature vs CS (below). The plots show that RH is as good as temperature under clear-sky conditions however it does not aid the separation (events from nonevents) further as CS sink seems to be the main controlling factor. We then conclude that during clear-sky conditions the results are somewhat different from what Hyvönen et al. 2005 who did not consider clear-sky conditions only. Based on the aforementioned results, and following the reviewer's suggestion, we add the following to the text to line 369:

*Furthermore, we analyzed the effect of RH in separating the events from nonevents, similar to the study done on RH by Hyvönen et al. 2005. However, when plotting CS vs RH (data not presented), our results show no enhanced separation of events from non-events based on RH when only clear-sky conditions are considered.*

[Figure]

**2.    Also, I think it should be made clear that the event probability described in Figure 13 and in section 3.3.4 is different from the equation 6, and also different from the event probability introduced in the Hyvänen et al paper. In the latter, the event probablility is computed from the LDA analysis, while in the current paper the probability seems to be directly calculated from data, and thus it is not a predictive equation. I suggest that the authors revise this part of the paper. Also, if no real propability-giving predictive equation is given, I think that aim IV in the Introduction (line 66) should be revised.**

As the reviewer mentioned, the event probability presented in figure 13 and in section 3.3.4 is calculated directly using the current data set. However, introducing such results best explains the direct effect of extreme temperatures and condensation sink on classifying days as events or non-events. The aim IV in the introduction, refers to equation 6 which sets the line for variable separation during clear-sky events and non-events. First, to improve our analysis, we perform LDA analysis to our dataset, similar to the analysis presented by Hyvönen et al. 2005. That made rather equation (6) more reasonable. Accordingly, equation (6) and the corresponding figures 11 and 12 are improved and replaced. Second, to make the aims clearer, following the reviewer's comments, aim IV in line 66 is divided into two to show the independence of equation (6) accompanied by figures 11 and 12 and the figure 13 modified to "*iv) formulate an equation that predicts whether a clear-sky day with specific temperature and CS is classified as an event; v) use the clear-sky data set to calculate the NPF probability distribution based on temperature and CS*".

**However, overall I think that the paper is a potentially good addition to the literature of understanding NPF, and its topic is certainly appropriate for ACP. Therefore, if the above corrections and the detailed notes given below can be considered by the authors, I would suggest publication. The corrections and revisions are, in my opinion, minor.**

**II.    Detailed notes:**

**1.    line 150-158 and 223-225: I do not fully understand the definition of the clear-sky day presented by the authors. Generally, it is known that particle formation occurs around noon, and that especially the mixing of the residual layer in the morning seems to have an influence. From that, I can follow that using the morning value is useful in the analysis. However, only the median P value for three hours is used. This raises the following questions: *i) Were only events that started during this three-hour window included in the analysis? *ii) Why was the median used? In this case, a time period that is for example 1 hour 29 minutes cloudy and 1 hour 31 minutes sunny gets classified as a sunny (clear-sky) day. Does the result change when the mean is used? *iii) what is the basis of using the value 0.7?**

 i)    NPF occurs usually in the morning hours and peaks around noon (Dada et. al 2017, In Preparation). For that, favorable conditions for clustering should be available to initiate the process as well as ensure its continuation. Since the sun-cycle varies widely in Hyytiälä between seasons which affects the NPF cycle also, the time window 9:00-12:00 seems to cover all seasons equally. Also, although all events are included in this classification, the ones that occur outside our selected time window are rather very few. For consistency, the variables compared in our study are taken between the same time window 9:00-12:00. Also, we do not aim at studying exhaustively the day-by-day results, rather formulate a picture on the variables that are very different on event days in comparison to non-events. And since up to our knowledge, no clear method has been detected to identify the start time of an event, doing the data mining manually for event days of such a characteristic (start time between 9:00 and 12:00) is very time consuming and adds no heavy value for the current paper.

ii)       We agree with the reviewer that it is tricky to define the day as clear/cloudy  period by a median or a mean value; however, let's assume that 2 hours have P = 0.6 and the remaining 1 hour is P=0.9, this will result in a mean of 0.7, clear-sky day while this is not the reality. However, we recalculated the difference in frequency of event occurrence in case we choose the mean P value for these three hours instead of the median, the differences are acceptable demonstrated in the table below:

|  | **Median** | **Mean** | **Percentage difference** |
|---|---|---|---|
| **Clear-sky events and non- events** | 1106 | 1045 | 5.5% |

We consider this difference insignificant for our analysis. The median value is useful also because NPF is a regional-scale phenomenon, so for instance scattered clouds on an otherwise sunny day affecting the local radiation measurements (and leading to a momentarily drop in P) do not usually interrupt the regional NPF process. The histogram (Count Normalized) below also shows that within the clear-sky days, the P values calculated every half hour between 9:00 and 12:00 almost never reach a value below 0.3 (which is the threshold for complete sky cover). This result advocates the fact that if any clouds appear on a day classified as clear-sky they are mostly scattered and not thick.

[Figure]

iii)       The value of 0.7 is found from previous studies which are mentioned in line 155. This has proven to be a value that works for different seasons, and is strict enough to exclude all totally or partly cloudy days, but not to eliminate sunny days with occasional scattered clouds passing over the station.

**2.     The reasoning between this central points in the methodology should be explained in much more detail, as I expect that similar analyses will be performed in the future for other sites, and therefore the method should be as robustly implemented as possible. Also, can the authors give insight on how sensitive the method is on the limit value of P chosen?**

The sensitivity of the method on the limit P value is shown in Figure 1a which shows the variation of the fraction of days when using different P values. We added more details to the method section 2.2.2:

*In Hyytiälä, the great majority of NPF events are initiated during the morning hours after the sunrise, yet before the noon (Dada et. al 2017, In Preparation). Since the time of the sunrise varies widely in Hyytiälä between the different seasons, the time window 9:00-12:00 seems a reasonable compromise for considering whether NPF did occur or not. We found that NPF events occurring outside our selected time window were very few. Accordingly, in this work the days were classified as cloudy or clear-sky days based on the median value of P during 9:00-12:00 each day, corresponding to the time window for new particle formation. Clear-sky days were those with a median of P > 0.7 between 9:00 and 12:00 and are the focus of this study. The median value ensures that at least half of our selected time window is clear-sky while the rest can vary between clear-sky and minor scattered clouds. The median is useful also because NPF is a regional-scale phenomenon, so for instance scattered clouds on an otherwise sunny day affecting the local radiation measurements (and leading to a momentarily drop in P) do not usually interrupt the regional NPF process. Clear-sky days were those with a median of P > 0.7 between 9:00 and 12:00 and are the focus of this study. For consistency, the variables compared in our study are taken between the same time window 9:00-12:00.*

**3.     Line 198: ". . . radiation is essential for NPF as these events occur mainly during daylight hours." If radiation was essential, no NPF could be observed during nighttime. In the literature, several examples of NPF during nighttime can be found. Please rephrase.**

We modified to *"radiation seems essential for NPF at this site, as the events occur almost solely during daylight hours."*

**4.     line 200: is SA really the main component of freshly formed particles? If heteromolecular nucleation is the prevailing mechanism, the the organic compound is as important. Both are still likely to be formed photochemically, so I think that this sentence can be fixed by just by rewording (e.g. '..because the main components of freshly formed particles are likely formed photochemically. . .')**

As suggested by the reviewer, we did the change.

**5.     line 235-245: Please clarify also in the text and in the caption of Figure 4 that these results refer to clear-sky events only.**

Line 235 is modified to *"The springtime medians are percentiles of air-mass trajectories arriving at Hyytiälä during clear-sky NPF events and non-events…."*

**6.     Line 251-254: As the CS is highest for event days, but not so for non-event days, does the presented conclusion that the CS is the reason for the minimum in events in summer really follow? It seems to me that in summer, events may occur despite high CS, and the actual reason for non-events is not the inhibiting effect of CS. If the authors disagree, this could be clarified.**

After modifying figure 10 and the accompanying text based on both reviewers' suggestions (See comment 9 below), it appears clearer that in summer, the calculated formation rates are high also during nonevent days, yet an event is not happening. This might be explained by higher temperatures in summer which leaves the freshly formed clusters rather unstable.

**7.      Line 270: with monthly I think that the authors mean yearly**

The whole section is rearranged to fit both reviewers' suggestions.

**8.      Line 280-281, '. . .low or almost no correlation. . .' something seems to be missing in this sentence.**

Line 280 is modified to "*However, during non-event days, a positive correlation appears between RH and each of CO, SO$_2$ and NOx while the correlation between those seems to be absent during event days.*"

**9.      lines 331-350: I don't really understand what is shown in figure 10, and therefore also don't follow the explanation in this paragraph. What is meant by diurnal cycle here? By definition it means a repeating pattern that occurs every 24 hours, and I don't see how this could result in Figure 10. Please clarify and rewrite, or replace with the correct figure.**

Figure 10 is replaced with median diurnal cycles of J$_3$ and CS during different seasons. The diurnal time frame is limited to 5:00-20:00 due to the incapability of calculating SA concentrations in the absence of UVB, therefore no J$_{3,C}$ values are calculated outside this time window.

**10.     Line 357: The procedure of finding the separating line in Fig 11-12 is described very poorly. Is this done by linear discriminant analysis (such as e.g. in the cited Hyvönen et al., (2005) paper or some other method? The authors should describe this in more detail. I'm especially concerned about the sentence "the data points have been estimated by taking the non-events with the lowest possible CS which still fit the linear ˇseparation"; was some kind of data selection applied to produce the figure?**

See comment 2 in the General comments section

**11.     Figures: Several figures have the sentence "The lines extending 1.5 times from the central box represent the remaining of the data yet still within the relevant statistical limit. " Please clarify what this means: firstly, what is 1.5 times from the central box (the lines seem to have different lengths, eg. in fig. 5. Also, clarify what is meant by relevant statistical limit.**

The extended lines are equal to 1.5 x interquartile range, and the points beyond whiskers are outliers (> 1.5 x interquartile range). Outliers are the individual stars. Maximum length of the extended line, is defined as q3 + w × (q3 – q1) while the minimum is q1 – w × (q3 – q1) where w, q1 and q3 are the mean, the 25th and 75th percentiles of the sample data, respectively. The statistical limit is defined by default as the 99.3% coverage. Based on the reviewer's comment we modify the text corresponding to the box plots for clarity to the following:

*The length of the whiskers represent 1.5 x interquartile range which includes 99.3% of the data. Data outside the whiskers are considered outliers and are marked with red crosses.*

**References**

L. Dada, R. Chellapermal, S. Buenrostro Mazon, V.M. Kerminen, P. Paasonen And M. Kulmala (2017). Method for identifying NPF event start and end times as well as NPF types (ion-initiated, particle initiated, transported..) using characteristic nucleation-mode particles and air ions. *In Preparation.*